# Learnable Stability-Aware Unstructured Grid Coarsening Using Graph Neural Networks for Accelerated Physics Simulations

## Abstract

Efficient simulations of complex physical systems described by partial differential equations (PDE) require computational methods that can reduce the resource demands without sacrificing the accuracy. Traditionally, this is achieved by "upscaling" the simulation grids or by aggregating cells based on a priori information. Here, we introduce a novel framework based on graph neural networks (GNN) for learnable self-supervised differentiable coarsening of unstructured computational grids. We leverage graph-based representation of the physical system and offer a graph coarsening method which preserves the underlying physical properties together with the stability of the chosen numerical scheme. This is achieved by minimizing the error between the output of the simulations using coarsened and original graph. We demonstrate the approach on several example differential equations, modeling sub-surface flow and wave propagation. We demonstrate that the model exhibits ability to maintain high fidelity in simulation outputs even after 95% reduction on the nodes, significantly reducing computational overhead. We also show that the model exhibits generalizability to unseen scenarios, thereby outperforming the baselines. Thus, the developed approach demonstrates the ability to accelerate simulation without comprising accuracy and hence has potential for accelerating physical simulations in various domains.

## 1 Introduction

Modeling fluid dynamics, particularly subsurface flow, remains a highly resource-intensive task due to the detailed spatio-temporal resolutions required for accurate simulations, as well as the non-linearity of the governing partial differential equations (PDEs) Gerritsen & Durlofsky (2005). Such simulations are crucial for numerous engineering applications; however, their computational demands often become a bottleneck, especially when multiple parameter variations are explored Huyakorn (2012). Therefore, reducing computational cost without sacrificing accuracy is a critical challenge in the field.

Traditionally, various methods, such as *upscaling*, have been employed to address this issue by replacing fine grids with coarser ones, while adjusting equation coefficients (e.g., permeability, porosity) to reflect the larger cells Qi & Hesketh (2005). While effective at reducing grid size, these methods typically assume a homogeneous approximation within each coarse block, limiting the ability to fully capture localized heterogeneity. More sophisticated upscaling methods solve PDEs within each coarse block, yet the overarching principle remains: reducing grid size to save computational time. Zhang et al. (2021); Farmer (2002)

Alternatively, *reduced-order modeling* (ROM) approaches focus on leveraging a subset of simulated data to accelerate the remaining computation. One notable example is proper orthogonal decomposition (POD), which utilizes dominant spatial eigenvectors for data approximation, akin to principle component analysis Kerschen et al. (2005); Rowley (2005); Guo & Hesthaven (2019). Dynamic mode decomposition (DMD) Kutz et al. (2016) is another recent method, utilizing both spatial and temporal frequencies to model and predict fluid dynamics. This approach has been extended further by learning operators that model the PDEs in the functional space Li et al. (2020); Kovachki et al. (2021; 2023). While they have been effective for wide range of problems, the absence of a physics-

based solver lead to: (i) divergence of errors for longer scale simulations, (ii) violation of physical laws such as conservation of energy and momenta, and (iii) lack of interpretability of the learned function. Burark et al. (2024); Azizzadenesheli et al. (2024)

*Differentiable physics* has emerged as a powerful framework for simulating and optimizing physical systems by incorporating gradient-based methods into the simulation process. The key advantage of this approach is that it allows for automatic differentiation (AD) at every step of the simulation Liang & Lin (2020), making it possible to compute the partial derivatives of the output with respect to the input. This has found applications not only in fluid dynamics but also in areas such as molecular dynamics for optimizing particle positions or force fields Schoenholz & Cubuk (2020); Gangan et al. (2024), and material structure design Dold & van Egmond (2023). However, integrating AD into PDE solvers, where the current state depends on previous states, creates highly nested computational graphs, which poses significant challenges for optimization Holl et al. (2020).

Recently, Shumilin et al. introduced an approach that employed grid coarsening for accelerated simulations through differentiable physics—the simulation and grid coarsening processes were made fully differentiable, including the finite-volume solver. This allowed to optimize the placement of points on unstructured grids by minimizing the misfit between coarse and fine grid simulations, leveraging techniques such as automatic differentiation (AD), $k$-means clustering and differentiable Voronoi tesselation. The method demonstrated the potential to reduce grid size while maintaining simulation quality, tested across various PDEs, including parabolic and hyperbolic equations.

Building upon this foundation, we now propose a framework based on graph neural networks (GNN) and graph pooling, which shifts the focus from optimizing point positions to a *self-supervised* learnable coarsening process. In contrast to the methodology proposed by Shumilin et al., where point positions were optimized for each specific grid, the GNN-based approach learns a generalizable coarsening strategy that can be applied to grids with varying numbers of points. This flexibility allows the coarsening procedure to adapt to different simulation setups without the need for grid-specific re-optimization, thus enhancing the scalability and applicability of the method.

The major contributions are as follows.

1. **Learnable graph coarsening:** We present a learnable, self-supervised GNN-based coarsening framework. It allows to learn optimal coarsening strategies for the grids of different sizes.
2. **Physics and stability losses:** We use physics loss to ensure that the coarsened grid yields the simualtions close to that of the ground truth respecting the governing PDE. We also propose a built-in stability loss for maintaining stability of numerical scheme used for explicit solver.
3. **Differentiable implicit numerical solver** We develop the differentiable implicit numerical finite volume solver that leverages differentiable Voronoi tesselation and extends the practical applicability of the proposed framework.

## 2 PRELIMINARIES AND PROBLEM FORMULATION

In this section, we formally introduce the concepts central to our work and formulate the problem of graph coarsening for physical simulations.

### 2.1 AUTODIFFERENTIATION

Automatic differentiation (AD) is a method for computing the gradients of a program's output with respect to its inputs Naumann (2012). Specifically, the computations are deconstructed into functions at intermediate steps generating a computational graph to which the chain rule is applied to obtain the instantaneous gradient. A key algorithm that utilizes AD is backpropagation, which efficiently calculates the gradient of a loss function concerning the network's weights. This significantly simplifies and enhances the learning process of neural networks.

Computational graphs are fundamental in various fields, including physics, chemistry, biology, and machine learning. They provide a structured representation of mathematical expressions and operations through nodes and edges. Each node corresponds to a specific operation, while the edges indicate the flow of data between nodes. This structure facilitates both forward and backward computation, which is essential for optimization tasks in neural networks. By representing complex functions as graphs, AD can efficiently compute gradients through backpropagation. There are a number of frameworks for AD such as JAX Bradbury et al. (2018b) and PyTorch (Paszke et al.,

2017). In our experiments, we use both PyTorch and JAX for explicit and implicit solvers, respectively, as discussed later.

## 2.2 GNNs for graph coarsening

We use the notation $\mathcal{G} = (\mathcal{V}, \mathcal{E}, A, X)$ to denote a graph over a finite, non-empty node set $\mathcal{V}$ and edge set $\mathcal{E} = \{(u, v) \mid u, v \in \mathcal{V}\}$. $A$ represents the adjacency (edge weight) matrix corresponding to the graph. $X \in \mathbb{R}^{|\mathcal{V}| \times d}$ denotes node attributes encoded using $d$-dimensional feature vectors. We denote the attributes of node $v$ as $\mathbf{x}_v$.

Given graph $\mathcal{G} = (\mathcal{V}, \mathcal{E}, A, X)$, graph coarsening is the process of constructing a significantly smaller graph $\tilde{\mathcal{G}} = (\tilde{\mathcal{V}}, \tilde{\mathcal{E}}, \tilde{A}, \tilde{X})$ with $|\tilde{\mathcal{V}}| \ll |\mathcal{V}|$ nodes, such that $\tilde{\mathcal{G}}$ and $\mathcal{G}$ have similar properties. This process involves merging multiple nodes into *supernodes* along with their aggregated features $\tilde{X}$. Hence, we need to learn a *surjective* mapping $\pi : \mathcal{V} \to \tilde{\mathcal{V}}$ that associates nodes in the original graph $\mathcal{G}$ to supernodes in $\tilde{\mathcal{G}}$.

In the context of grid coarsening for physical simulations, we assume the dataset to be a set of graph snapshots $\mathbb{G} = \{\mathcal{G}^1, \cdots, \mathcal{G}^T\}$, where each snapshot $\mathcal{G}_t \in \mathbb{G}$ shares the same topology, but potentially varying node attributes, i.e., $\mathcal{G}^t = (\mathcal{V}, \mathcal{E}, A, X^t)$. Furthermore, For each node $v \in \mathcal{V}$, we associate a time-dependent physical quantity $y_v^t$ (e.g., pressure) as ground truth, which is obtained from a physics-based simulation or experimental observations. The complete ground truth information is denoted as $Y = \{\mathbf{y}^1, \ldots, \mathbf{y}^T\}$, where $\mathbf{y}^t = \{y_v^t \mid \forall v \in \mathcal{V}\}$. However, when the input graph is large spanning thousands of nodes and edges, physics-based simulations can become prohibitively expensive. To address this, the GNN trained on $\mathbb{G}$ seeks to learn a coarsened graph along with their feature to model the evolutionary dynamics of $y_v^t$ as a function of the graph topology and node attributes. Thus, we propose to coarsen the topology of the input graph $\mathcal{G}$ to $\tilde{\mathcal{G}}$ such that the output of the physics-based simulation on $\tilde{\mathcal{G}}$ closely approximates the simulation output on the original graph $\mathcal{G}^1$. Formal definition of problem is presented in Appendix A.

**Related work**. Although there are numerous works investigating mesh coarsening, our approach is distinct in that it integrates an explicit and implicit numerical and differentiable finite volume solvers directly into the training loop without relying on surrogate models or approximations. For instance, Graph Element Networks (GENs) introduced by Alet et al. (2019) approximate the solution of PDEs using GNNs trained on high-resolution solutions. GENs adjust node positions or densities to better capture the solution space but do not perform explicit graph coarsening through node aggregation or clustering. Additionally, GENs focus on stationary PDEs or displacement prediction and do not incorporate time dependence in their implementation, limiting their applicability to dynamic simulations. Subsequently, Pfaff et al. (2020) proposed MeshGraphNets, which apply remeshing techniques that may increase the number of cells and computational load rather than coarsening the mesh. Their remesher coarsens the mesh by removing edges only when it does not create invalid elements, based on a purely geometrical criterion involving the sizing field tensor predicted by the model. This approach is primarily applied to 2D manifolds embedded in 3D space, such as flag or sphere scenarios, and relies on local curvature. In contrast, our method utilizes simulation data misfit to cluster and aggregate nodes, resulting in a more physically informed coarsening process applicable to general graphs. More recently, Li et al. (2024) developed a differentiable finite volume method using GNNs within an encoder-decoder framework and employed a PDE loss for training. However, their solver functions as a surrogate model, as they do not integrate the actual physics solver into the optimization process. In our work, we incorporate explicit and implicit numerical differentiable non-surrogate finite volume solvers directly into the training loop, enabling end-to-end learning of coarsened graphs that accurately capture the dynamics of the original system. This approach broadens the applicability of graph coarsening methods in physics-based simulations, particularly for time-dependent and stiff PDEs.

In our approach we use a Graph Convolutional Network (Kipf & Welling, 2017) for producing a learnable coarsening, its architecture is defined in Section 3.2.

---

[1]Since the topology is static, we ignore the timestamp when the discussion is centered on coarsening

## 3 PROPOSED METHODOLOGY

To demonstrate the applicability of our graph coarsening approach, we consider for several PDEs and both the explicit and implicit solvers. We implement the solvers for the equation in both PyTorch Paszke et al. (2019) and JAX Bradbury et al. (2018a). These implementation allow us to build the computational graph and backpropagate through the physics calculations. To implement explicit solvers we use PyTorch Geometric library Fey & Lenssen (2019) with Message Passing interface.

### 3.1 EQUATIONS AND DIFFERENTIABLE SOLVERS

**Diffusion equation.** The first example we consider is the diffusion equation as discussed below. Let $V$ be a polygonal domain. In general, the diffusion equation takes the following form:

$$\frac{\partial u}{\partial t} - \text{div}(K \nabla u) = f, \quad 0 < t < T, \tag{1}$$

where $u(x, y, t)$ is the unknown pressure, $K(x, y)$ is the function for diffusion coefficient, $f(x, y, t)$ is the source/sink term. The above equation is completed with the initial condition $u = u^0(x, y)$ at $t = 0$ and zero Neumann boundary conditions on the domain boundary.

**Explicit Euler scheme**. To obtain a numerical solution of Equation 1, we apply the finite volume method for spatial discretization Eymard et al. (2000); Kuznetsov et al. (2007) and the forward Euler scheme for temporal discretization. Let $m$ be the number of time steps of size $\tau = T/m$ and $S_i$, $i = 1 \cdots N$, be arbitrary points in $V$. We form a Voronoi mesh using these points, and denote as $V_i$ the respective cells, $|V_i|$ its area, as $e_{ij}$ the edge separating adjacent cells $V_i$ and $V_j$, $|e_{ij}|$ its length, $h_{ij}$ distance between Voronoi cites of two adjacent cells $V_i$ and $V_j$.

Now, the discretization of Equation 1 reads as follows,

$$D \frac{u^{k+1} - u^k}{\tau} + Au^k = Df^k, \quad k = 0 \cdots m - 1, \tag{2}$$

where $u^k \in \mathbb{R}^N$ and $f^k \in \mathbb{R}^N$ are discrete pressure and source vectors, respectively, $D$ is a diagonal matrix of areas $|V_i|$ and $A \in \mathbb{R}^{N \times N}$ is the finite-volume system matrix. It is a sparse symmetric matrix with the following entries. If cells $V_i$ and $V_j$ are adjacent, then

$$A_{ij} = -\frac{|e_{ij}|}{h_{ij}} \frac{2K_j K_j}{K_i + K_j},$$

otherwise off-diagonal entries are zero, and the diagonal entry is equal to the negative sum of off-diagonal entries.

$$A_{ii} = - \sum_{j=1, j \neq i}^{N} A_{ij}.$$

Since the expressions for matrix entries are fairly simple we do not actually store the matrix, but rather we compute the product $Ap^k$ on-the-fly.

**Implicit Euler scheme**. The cell sizes in our method change dynamically and sometime cells with a small areas emerge. This may cause the stability issues. To handle this we implement implicit solvers known for their stability compared to the explicit ones. In case of implicit Euler scheme, we use backward difference for $\partial u / \partial t$. The equation 1 takes the form:

$$D \frac{u^k - u^{k-1}}{\tau} + Au^k = Df^k, \quad k = 0 \cdots m - 1. \tag{3}$$

We can only express $u^k$ implicitly by a matrix equation. The desired solution $u^k$ is obtained by solving a system of linear equations (SLE):

$$[D + \tau A] u^k = Du^{k-1} + \tau Df^k, \quad k = 0 \cdots m - 1. \tag{4}$$

Traditionally standard SLE solvers are applied to solve Equation 4. However, standard SLE solvers does not allow automatic differentiation by passing gradients. To incorporate a differentiable version of SLE solver into our pipeline, we use JAX programming library. As a method of choice,

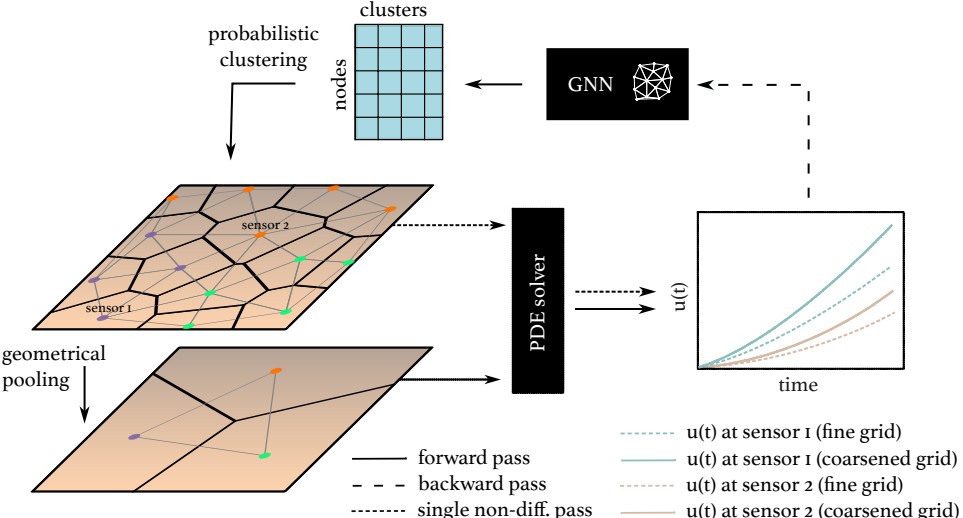

Figure 1: The proposed graph coarsening framework. Graph convolution (represented by GNN) predicts the cluster assignments which is used to aggregate the features and obtain a coarsened grid. The GNN is trained by minimizing the error between the field quantity obtained from the simulations using the coarse and the fine (ground truth) grids.

we leverage a sparse version of QR factorization because the matrix $A$ is sparse. The sparsity of $A$ is supported by a property of the underlying Voronoi tessellation: for a cell in a Voronoi tessellation the average number of neighbors is less than 6 (Lemma 2.3 from Aurenhammer et al. (2013)). This statement implies that the matrix $A$ has less than 6 plus one non-zero entries on average.

**Wave equation**. As a second example, we consider the simulation of the process of propagation of sound waves in fluids, described by a linearized hyperbolic equation. We omit viscous forces, temperature effects, and body forces. Finally, we get the standard linear wave equation for pressure $p$:

$$\frac{\partial^2 u}{\partial t^2} = c^2 \nabla^2 u,$$

where $c$ is the speed of sound in the fluid, we accept $c = 1$ for simplicity. Using previously defined notations for the elements of Voronoi tessellation, we apply the second order temporal discretization combined with the finite volume spatial discretization:

$$D\frac{u^{k+1} - 2u^k + u^{k-1}}{\tau^2} + Au^k = Df^k, \quad k = 0 \cdots m - 1. \tag{5}$$

Note that the matrix $A$ in Eq. 5 is the same as that in Eq. 2. Euler explicit scheme is typically used for the wave equation. Combining these equations and leveraging the forward Euler scheme we get the following expression for $u^{k+1}$:

$$u^{k+1} = -u^{k-1} + 2u^k + D^{-1}\tau^2 Au^k + \tau^2 f^k$$

To solve the second order equation, we apply two initial conditions: $u(x,0) = 0$, and $\frac{\partial u}{\partial t} = \psi(x)$. Further, we apply free boundary condition that allows pressure waves to move through the boundary.

## 3.2 GRAPH COARSENING PIPELINE

The overall framework employed in the present work is presented in Fig. 1. Our NN architecture for coarsening consists of a one graph convolution layer followed by a point-wise MLP with two fully connected layers and softmax at the end, which predicts the soft assignment matrix $S_{ij}$ for probabilistic clustering. These assignments are used to aggregate node features, ensuring that the key physical properties of the system are preserved in the coarsened representation. Notably, the features of sources and sinks are kept intact during the coarsening process to maintain the integrity of the flow dynamics. We set which nodes to keep as a hyperparameter. The motivation for this problem comes from the field of proxy modeling and other engineering fields where model identification needs to be performed based on limited sensor measurements. See Appendix C for more details. Following this physics-based simulation is performed using the PDE solver to obtain the output

quantities based on the coarsened graph. To ensure numerical stability, we utilize max subtraction for softmax stabilization. Finally, the model is trained by minimizing the error between the output quantity obtained from the coarsened graph-based simulations with respect to those obtained from the original simulations.

The GNN model for coarsening is trained using a two-stage loss function that balances stability and physical fidelity (detailed in Sec. 3.4). For the latter, we use the so-called physics loss, which compares the root mean squared error (RMSE) between the simulated dynamics on the coarsened graph and the ground-truth fine-mesh simulation over time. The RMSE is computed based on the time series data of key variables in the simulation.

When using an explicit solver, a warm-up period of 50 epochs is used by default, where only the stability loss is optimized. This stability loss leads to maximizing permissible timestep that ensures the numerical scheme remains stable and does not diverge (see mathematical definition 8 below). After the warm-up, the algorithm checks the stability condition described in Section 3.4 at each step. If the condition is satisfied, both physics and stability losses are used for optimization. If not, only the stability loss is applied. However, when an implicit solver is used, only the physics loss is applied, as implicit schemes inherently have good stability properties. Additionally, gradient clipping is applied to prevent exploding gradients, and the adjacency matrix and geometry are updated dynamically after each iteration to reflect the new node positions. All the codes used in the present work are available in: `https://anonymous.4open.science/r/Learnable-coarsening-9434`. Currently, our framework is designed for 2D computational domains. The extension to 3D differentiable Voronoi tessellations and handling of non-simplex grids is discussed in Appendix B.

**Source and sink treatment in fluid flow simulations.** In subsurface flow simulations, fluid movement is primarily driven by wells that inject or extract fluids. These wells are often modeled as point sources where the intensity of the source depends on the pressure difference between the wellbore and the surrounding area. When discretized, the source term can be written as:

$$f_i^k = \begin{cases} c_\alpha(p_{bh,\alpha} - p_\alpha^k) & \text{if } i = \alpha, \\ 0 & \text{otherwise,} \end{cases}$$

where $\alpha$ represents the well's location, $c_\alpha$ is a coefficient depending on factors such as well size and local permeability, and $p_{bh,\alpha}$ is the pressure at the wellbore. As mentioned earlier, the source and sink nodes are kept intact in our coarsening scheme. Further, the pressure values at these nodes are used as the loss function for training the coarsening model (see Sec. 3.4).

### 3.3 FEATURE AGGREGATION

Note that an important aspect of coarsening is to learn the features of the coarsened graph. To this extent, our learnable coarsening method aligns with the cluster centroid calculation scheme in soft $k$-means clustering Dunn (1973); Bezdek (2013). In soft $k$-means, each point $\mathbf{x}_i$ is assigned to cluster $j$ based on probabilistic membership. The cluster centroid $\mathbf{c}_j$ is then updated as a weighted average:

$$\mathbf{c}_j = \frac{\sum_{i=1}^N S_{ij}\mathbf{x}_i}{\sum_{i=1}^N S_{ij}}$$

In our GNN-based approach, the soft assignment matrix $S$ is learned directly by the neural network, allowing $S_{ij}$ to adapt dynamically based on both the graph structure and physical properties. In our case, the feature vector $\mathbf{x}_i$ consists of the spatial coordinates $x$, $y$, and permeability $k$, ensuring that the resulting coarse grid retains both geometric and physics information from the fine grid. Furthermore, this aggregation scheme preserves the range of the original coordinates and permeability, while maintaining the physical realism of the model.

### 3.4 LOSS FUNCTION

Our loss function is an aggregation of *physics loss* and *stability loss*. During optimization, we utilize a multi-step approach where gradients are accumulated and updated across the entire simulation rollout in time.

**Physics Loss:** Let $p_s(t)$ and $p_s^*(t)$ be the time series for modelled variable (in our case, pressure) at a sensor point $s$ modeled on initial grid $S$ and coarsened grid $S^*$. The sensor point notation is the same on both grids because these points remain untouched by the coarsening procedure. Let $M$ be the number of sinks and $T$ be the number of timesteps. We define the physics loss based on the root mean squared error (RMSE) between the pressure values simulated on both grids.

$$\mathcal{L}_{physics} = \frac{1}{T} \sum_{t=1}^{T} \sqrt{\frac{1}{M} \sum_{s=1}^{M} \left( p_s(t) - p_s^*(t) \right)^2} \tag{6}$$

**Stability loss and stability of the numerical scheme:** The sufficient condition for stability of the forward Euler scheme takes the following form,

$$\tau \leq \frac{V_{min}}{C \, K_{max}}. \tag{7}$$

where $V_{min}$ is the smallest cell area, $K_{max}$ is the largest permeability, and $C$ is a dimensionless constant, which depends on the geometry and topology of the grid. It can be further shown that $C \geq 4$.

The stability inequality motivates us to introduce a stability loss, a loss function that promotes grids with weaker restrictions on $\tau$,

$$\mathcal{L}_{stability} = -\frac{4}{C} \, \frac{nV_{min}}{|V|}. \tag{8}$$

We added the negative sign to the right-hand side of equation 7 and made the factors dimensionless. We neglected the contribution of $K$ since the maximum permeability does not change during coarsening. Also notice $4/C \leq 1$ and $nV_{min}/|V| \leq 1$. In our implementation, the actual stability loss is computed as a sigmoid function applied to the scaled stability loss: `sigmoid_weight` $\times$ `stability_loss`. This ensures that the stability loss is bounded between 0 and 1, preventing extreme gradients during optimization. The `sigmoid_weight` parameter controls the strength of the stability constraint, balancing its influence in the overall optimization.

A geometrical interpretation could be given to this loss function: it discourages optimization to design pathological grids, e.g. grids where one cell has too many neighbors or grids where cells' areas differ significantly.

## 4 Experiments and Analysis

Here, we consider several scenarios for evaluating our coarsening pipeline as outlined below. Note that the visualization of the permeability fields in the datasets and the details of the hyper-parameters are provided in Appendices D and I, respectively.

### 4.1 Baselines

Note that machine learning based grid coarsening for PDEs, being a recent approach, has limited number of existing methods. To evaluate our approach, we use two baselines described in Shumilin et al., as follows.

- $k$**-means + averaging**, consisting of $k$-means clustering of the nodes and averaging of the permeabilities and node positions within the clusters;
- **Shumilin et al.**, which is based on the global optimization of the nodes positions referred further in text as the competing approach.
- **METIS** Karypis & Kumar (1998), well-known software tool designed for partitioning graphs and meshes. METIS aims to generate blocks that are uniform in size and shape, minimizing the number of connections between different partitions.

It is worth noting that the competing approach has one major drawback—the necessity of costly global optimization in every modeling task. We hypothesize that: (i) the competing approach may yield better results but much worse generalizability, as the present approach may be applied in inference mode to unseen data under some limitations, (ii) our approach may produce even better results in some cases due to more flexible averaging procedure. These are explored in the experiments later.

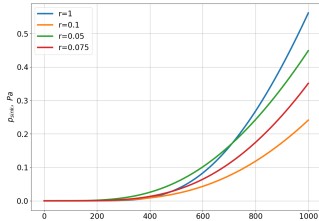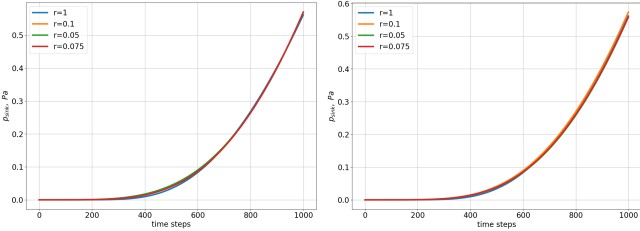

Figure 3: Comparison of $p_s$ for different degrees of reduction for *loop* scenario for: a) $k$-means + averaging approach. b) method from Shumilin et al.. c) our method.

## 4.2 "LOOP" SCENARIO

As a first scenario, we use the configuration **loop**, as in Shumilin et al.. This scenario represents an evenly spaced point cloud $S$ ($N = 312$) where the discrete permeability field $K$ has values 0.1, 1 (see Appendix D). For the simulations, we use the following parameters: $m = 10^3$, $\tau = 10^{-4}$, $p_{bh,src} = 100 \, Pa$, $c_{src} = 1.0 m^3/s$. Then, we compare $p_{sink}$ for $0 < t < T$ for the four scenarios: $r = \{1.0, 0.10, 0.075, 0.05\}$. We set a $[0, 1]^2$ boundary.

Note that in the present experiment, we choose different degrees of reduction, defined as $r = n/N$, to coarsen this grid. Here, $n$ and $N$ represents the number of nodes in the coarsened and original grid, respectively. An $r$-value of 1 refers to the ground truth grid. Then, the coarsened grids are used with the explicit FV solver to obtain the simulated $p(x, y, t)$ at the sink point. This results in a pressure series vector $p_s \in \mathbb{R}^m$. The performance is compared with method from Shumilin et al., which optimizes the locations of grid points, and $k$-means + averaging as described in the Sec. 4.1. The results of the experiment are shown in Fig. 3. The results show that our method has a quality comparable to that of Shumilin et al.. The comparison of RMSE for this experiment is presented on the Fig. 2.

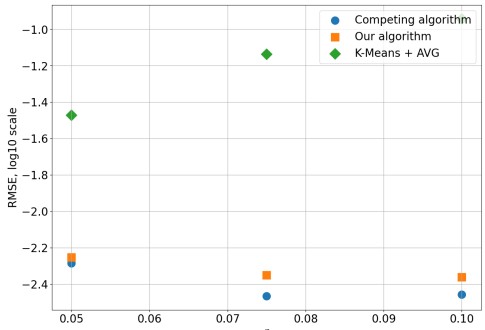

Figure 2: The result shows our method is comparable to Shumilin et al.. This allows us to coarsen without compromising the modeling quality.

We also use a comparison with the coarsening algorithm from METIS. If we let fine-scale transmissibilities measure connection strengths between cells for the edge-cut minimization algorithm, the software tries to construct a block without crossing large permeability contrasts. The results of the experiment demonstrated in Appendix K It can be seen from the results that the result obtained by our algorithm is much more accurate than when using METIS.

For many industry tasks it is significantly important to measure how well model predicts future on which it has not been trained on. We test whether our pipeline can output a proxy model that is capable of predicting the future oil debit. See Appendix J for details and results of this experiment.

## 4.3 SINUSOIDAL VARIABLE PERMEABILITY SCENARIO

In this subsection, we evaluate the performance of our algorithm using a synthetic *sinusoidal variable permeability* field (See Appendix D). This scenario tests the adaptability and efficiency of our method in handling spatially varying permeability fields, which are common in geophysical and environmental simulations.

The 2D domain is set to a unit square with side lengths of $l = 1$. The grid is initialized with a resolution of $20 \times 20$ points, resulting in a total of $400$ grid points. Each grid point has coordinates $(x, y)$, and the permeability $K(x, y)$ is generated using a sinusoidal function that varies spatially across the grid. Specifically, the permeability field $k(x, y)$ is defined as:

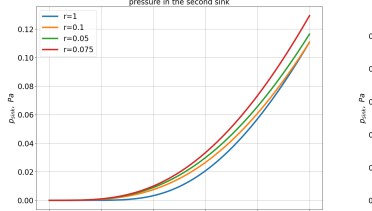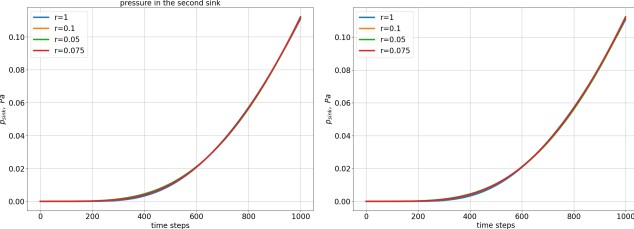

Figure 4: Comparison of $p_s$ in the first sink for different degrees of reduction for *sinusoidal variable permeability* scenario for: a) $k$-means + averaging approach. b) method from Shumilin et al.. c) our method.

$$K(x, y) = \cos(20x) + \sin(y) + \text{shift}$$

To ensure that all values of $K(x, y)$ are positive, we add a constant shift shift $= 2.0 + \text{abs}(\min(K(x, y)))$. This guarantees that the permeability values remain physically meaningful throughout the domain.

We simulate this permeability field with sources and sinks as defined:

- A single source is placed at grid point (0.53, 0.53) with a constant injection rate of 100 units.
- Two sinks are located at grid points (0.1, 0.1) and (0.89, 0.89).

In that case, we have $\tau = 10^{-5}$ and the parameters: $m$, $p_{bh,src}$, $c_{src}$ are identical to the "loop" case. The boundary conditions are set slightly outside the unit square to allow for smoother boundary processing in the coarsening methods. We evaluated three methods on the sinusoidal variable permeability scenario, each tasked with performing grid coarsening at various degrees of reduction (10%, 7.5% and 5% of the original grid resolution). We compared the performance of the three methods at various levels of grid reduction. The following figures demonstrate the coarsened grids produced by each method and the corresponding simulation accuracy: Fig. 4.

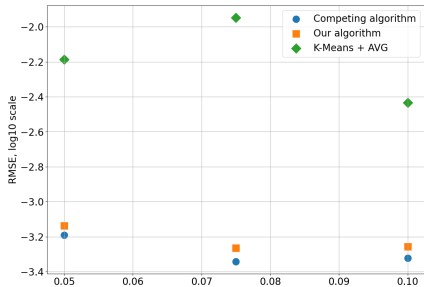

Figure 5: Our algorithm achieves comparable results with algorithm by Shumilin et al. and better than $k$-means

In a sinusoidal variable permeability scenario, our proposed method has comparable performance to the point optimization method demonstrated in Shumilin et al., while surpassing the basic $k$-means method in terms of maintaining high modeling accuracy. This performance is demonstrated in Fig. 5. The comparison with Metis demonstrated in Appendix K. In addition, as for the "loop" scenario our algorithm has better performance.

### 4.4 USING TRAINED MODEL ON DIFFERENT CONFIGURATIONS

Now let's consider the loop scenario and taking the model parameters described in Section 4.2. For performing grid coarsening at a reduction degree of 7.5%, we train our coarsening algorithm. Next, we take the original permeability distribution field containing 312 points and find the coarsened point cloud using $k$-means + averaging approach, the resulting values serve as the initial permeability field for future comparisons(experiments were conducted for grids with 200, 150, and 100 points). We then apply the coarsening method presented in this article and obtain results for pressure. The results of the experiments are demonstrated on the Fig. 9 in Appendix F.

Now we take our trained model for the degree of coarsening 7.5% described in Section 4.2. After that, we apply our trained coarsening model for the initial permeability field but change the coordinates for the sinks. Our first sink located at grid point (0.48, 0.58), second at grid point (0.39, 0.5), the third sink at (0.22, 0.33) and as we know original sink is located at (0.52, 0.58). See Fig. 7 in Appendix E. The result of measuring pressure at the sink point demonstrated on the Fig. 8 in Appendix F.

The results indicate that when the new sinks are positioned near the original sink, we achieve accurate pressure results on the coarsened grid (results for the first and second sink on Fig. 8 in Appendix F), but in the case of choosing a third sink that is located far from the original, our result becomes worse.

These findings demonstrate that our proposed coarsening method after training can be used in other initial simulation conditions and this does not require training the coarsening algorithm again, which proves its better generalizability in comparison for example with Shumilin et al..

### 4.5 ADDITIONAL EXPERIMENTS

**Log-uniform permeability scenario**

Now we consider an experiment with a significantly more complex permeability. We have 2D domain is set to a unit square with side lengths of $l = 1$. In this experiment, we generate permeability based on a log-uniform distribution. See Appendix L for details and results.

**Implicit Solver.** Another way to improve the stability of our method is to use an implicit solver as described in Section 3.1 for the diffusion equation. The solver is implemented in the open-source code and the demonstration of its work is available here[1] . Please read Appendix M for more information, including motivation of using implicit solvers.

**Wave equation**. We also verified the applicability of the proposed scheme on the wave equation (hyperbolic). Because of the reduced stability restrictions the scheme worked well without the stability loss. The demonstration is also available in the same folder[1].

**Time and memory tests**. Details of the time and memory performance results are available in Appendix H.

## 5 CONCLUSION

In this work, we introduce a novel flexible framework for unstructured grid coarsening based on GNNs, advancing the state of grid optimization from position-based adjustments to a learnable, self-supervised coarsening process. Importantly, our pipeline is physically motivated and aggregates features in physically correct way also taking care of the numerical scheme stability. Our approach can be used on different grid sizes, eliminating the need for grid-specific re-optimization, making it scalable for a wide range of simulation setups.

Our experiments demonstrate that the present approach delivers comparable accuracy to the work of Shumilin et al. while giving a better generalizability. We validated this in the sinusoidal variable permeability scenario, where the results closely match the fine-mesh simulations, showcasing the robustness of our method. Furthermore, the adaptability of the learned model to grids of different sizes and another initial conditions without retraining provides a clear advantage over previous methods, including the work of Shumilin et al., where costly optimization is required for every new grid. This capacity for generalization paves the way for efficient simulations with reduced computational costs and without sacrificing accuracy. We also show the applicability of our method for the hyperbolic wave equation here[1]. Due to good stability properties of an explicit scheme for the wave equation we are able to optimize physical loss only. Overall, the framework we propose opens new possibilities for scalable and adaptable physics simulations, reducing both computational resources and the environmental footprint associated with large-scale simulations.

**Limitations and future work.** The present work while promising is still applied to only two types of equations (parabolic and hyperbolic, describing subsurface flows and waves). The applicability of the approach to more complex flows and other physics equations is an open problem. The approach still relies on differentiable physics-based solvers and hence has the limitations of the simulations themselves such as small timestep and stability to name a few. Thus, combining the present work with data-driven models or graph neural ODEs could be a promising future work. Finally, the present approach employs a simple message passing GNN. It would be interesting explore the performance of the framework on different graph architectures with inductive biases such as equivariant GNNs.

---

[1]https://drive.google.com/drive/folders/1Aygtfvio3JB8ZxK0HAU3jIpIQolKaAk-?usp=drive_link

## 6 Reproducibility

Link to anonymous repository containing code of experiments: `https://anonymous.4open.science/r/Learnable-coarsening-9434`. In Appendix I, we list the hyper-parameters used for the experiments. The hardware specifications of the machine where the experiments were conducted is outlined in Appendix H.

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

## A GRAPH COARSENING

**Problem 1** (Graph Coarsening for physics-based simulations). *Let* $\mathbb{G} = \{\mathcal{G}^1, \cdots, \mathcal{G}^T\}$ *be the input graphs with associated ground-truth data* $Y \in \mathbb{R}^{T \times |\mathcal{V}|}$. *The goal is to obtain coarsened graphs* $\tilde{\mathcal{G}}^t = (\tilde{\mathcal{V}}, \tilde{\mathcal{E}}, \tilde{A}, \tilde{X}^t)$ *from a* GNN *model* $\mathcal{M}(\tilde{\mathbb{G}}; \Theta)$, *parameterized by* $\Theta$ *and output from the physics-based simulation as* $\tilde{Y} = \{\tilde{\mathbf{y}}^1, \cdots, \tilde{\mathbf{y}}^T\}$, *that satisfies the following:*

$$\min_{\Theta} \mathcal{L}\left[\tilde{Y}, Y\right] \tag{9}$$

where $\mathcal{L}$ denotes the loss function. We also need to an devise aggregation function $f_x : (X^t, \pi) \to \tilde{X}^t$. The exact aggregation function $f_x$ and the loss function $\mathcal{L}\left[\tilde{Y}, Y\right]$ is defined in Section 3.3 and Section 3.4 respectively.

## B HANDLING MESH RECONSTRUCTION AFTER COARSENING

Currently, our framework operates in 2D, where we construct a new Delaunay triangulation followed by a differentiable Voronoi tessellation after obtaining the coarsened nodes. This process is efficient, with a computational complexity of $O(n \log n)$, where $n$ is the number of coarsened points, significantly smaller than the original number of nodes ($n \ll N$).

For 3D domains, the main adaptation involves replacing the 2D differentiable Voronoi tessellation with its 3D counterpart. We actively working on extending our pipeline to support 3D differentiable Voronoi setups. Importantly, our GNN and finite volume simulator are agnostic to the dimensionality of the computational domain, ensuring that these components will seamlessly adapt to 3D setups in the future.

In cases where the mesh is non-simplex, maintaining regularity post-coarsening can be challenging. To address this, we rely on the flexibility of the finite volume solver, which effectively handles irregular grids while maintaining the accuracy and stability of the simulation. Our framework is particularly well-suited for regular grids, where differentiable coarsening and pooling steps are naturally supported.

## C RATIONALE FOR PRESERVING KEY POINTS

Consider an oil reservoir with wells where measurements are available - we are only interested in simulating the pressure at these particular points. Thus, if we achieve speed-up of computation at these positions while preserving the physics, we consider the goal to be completed. Indeed, if higher accuracy is required in certain regions of interest, additional sensor points can be incorporated into these areas to reduce interpolation errors. It can be implemented by the a priori analysis of the initial data: we may compute the gradient field and choose points that have higher gradients as sensor points. Also, we may use other algorithms based on graph spectral decomposition and other importance metrics e.g. centrality measures. This adaptive selection of sensor points ensures that regions requiring more precise solutions receive adequate attention without significantly increasing the computational gains.

In our current implementation, we assume that sensor locations align with the grid points. However, we recognize that in some practical applications, this may not always be the case. To address scenarios where sensors do not coincide with grid nodes, we plan to incorporate interpolation techniques such as Barycentric interpolation Berrut & Trefethen (2004); Floater (2015) in future work. These methods allow us to estimate sensor values based on surrounding grid points while ensuring that the interpolation weights remain differentiable with respect to grid parameters. This approach would preserve the overall differentiability of our computational pipeline, maintaining compatibility with gradient-based optimization.

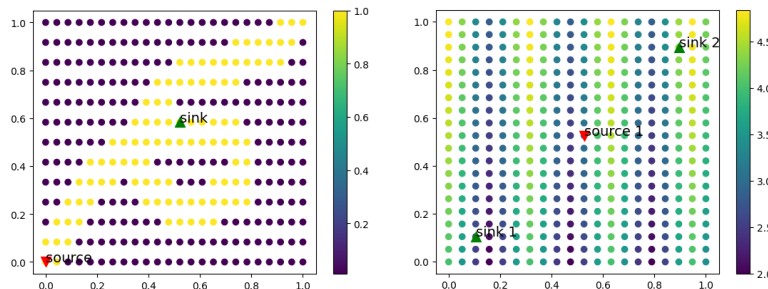

Figure 6: Permeability fields: a) "loop" scenario. b) sinusoidal variable permeability field.

## D    PERMEABILITY FIELDS USED IN OUR EXPERIMENTS

## E    COARSENED PERMEABILITY FIELDS FOR LOOP CASE FOR DIFFERENT POSITIONS OF SINKS

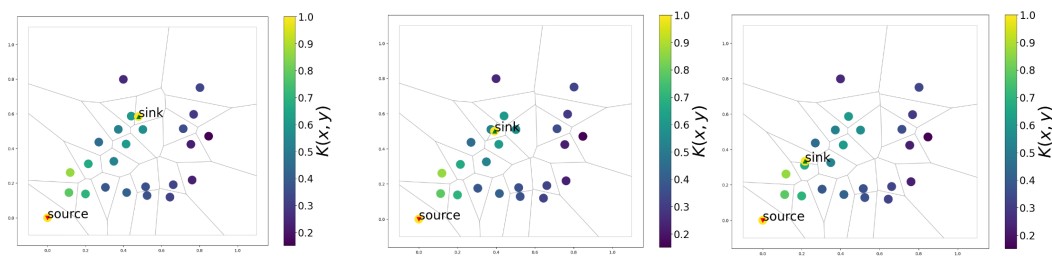

Figure 7: Point clouds after reduction: a) first sink, b) second sink, c) third sink.

## F    INFERENCE WITH DIFFERENT CONFIGURATIONS

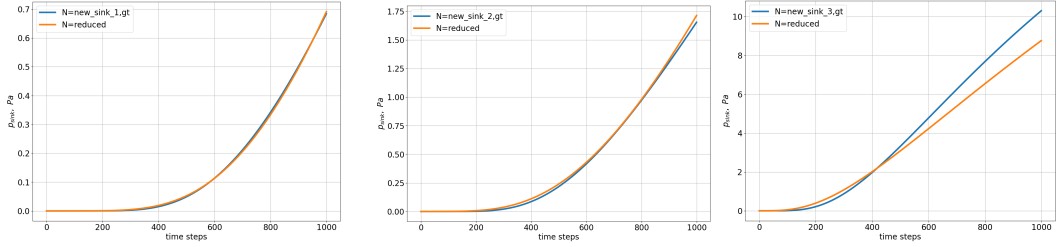

Figure 8: Comparison of $p_s$ for different locations of our new sinks: a) result for first sink, b) result for second sink, c) result for third sink.

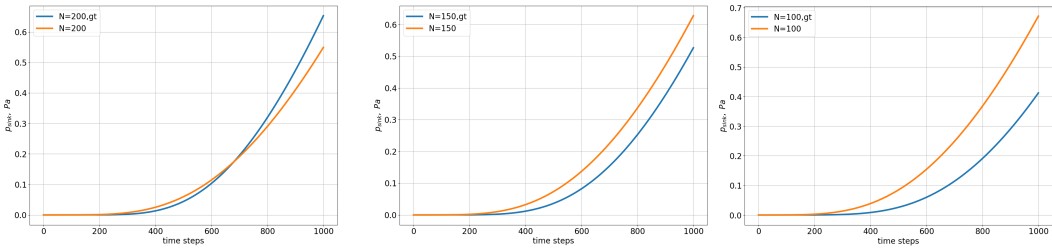

Figure 9: Comparison of $p_s$ for different cloud sizes for *loop* scenario: a trained model applied to other clouds with pre-applied kmeans.

## G  Log-uniform "Hard" permeability fields

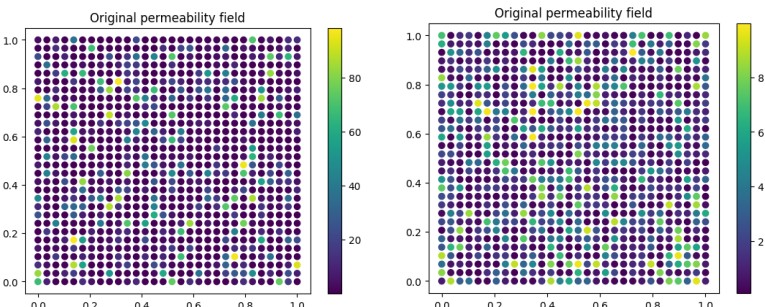

Figure 10: Permeability fields: a) when we keep values less than 100 and greater than 0.01 b) when we keep values less than 10 and greater than 0.1

## H  Results of time and memory tests

We compared the time performance of our method and method from Shumilin et al.. The experiments were conducted on Google Colab, using two cores of an Intel(R) Xeon(R) CPU @ 2.20GHz and 12.7 GB of RAM. The experiments were conducted on grids with a sinusoidal permeability field, varying between 2.5 and 4.5. All runs used 10 epochs of optimization and 100 timesteps. Our method, while slower on individual tasks, has the same order of time, especially for grids having less than 10000 points. However, our approach does not need new optimization for each new task.

Table 1: Time comparison between new and algorithm from Shumilin et al. (called "competitive" there)

| Grid Size | Algorithm (Clusters) | Time (s) |
|---|---|---|
| 4900 | New (50) | 8.5 ± 0.87 |
| 4900 | New (30) | 7.66 ± 0.85 |
| 4900 | Competitive (50) | 5.56 ± 1.14 |
| 4900 | Competitive (30) | 4.31 ± 0.51 |
| 8100 | New (50) | 7.9 ± 1.05 |
| 8100 | New (30) | 8.85 ± 2.39 |
| 8100 | Competitive (50) | 5.44 ± 0.44 |
| 8100 | Competitive (30) | 4.84 ± 0.62 |
| 22500 | New (50) | 26.1 ± 5.54 |
| 22500 | New (30) | 22.4 ± 0.44 |
| 22500 | Competitive (50) | 6.55 ± 1.31 |
| 22500 | Competitive (30) | 5.00 ± 0.43 |

Memory-wise, our method shows higher usage compared to the baseline. For example, in the 22500-point test with 50 clusters, our approach used around 6.5 GB, while the baseline consumed around 4.7 GB. Although our method requires more computation time and memory, it offers significant advantages in terms of flexibility. Once trained on diverse configurations, it can be applied to grids of varying sizes without the need for retraining, offering a solution to large-scale and dynamically changing simulation problems.

Table 2: Memory comparison between new and algorithm from Shumilin et al. (referred to as "competitive" in the work).

| Grid Size | Algorithm (Clusters) | Memory Peak (MiB) |
|---|---|---|
| 4900 | New (50) | 1069 |
| 4900 | New (30) | 1218 |
| 4900 | Competitive (50) | 888 |
| 4900 | Competitive (30) | 984 |
| 8100 | New (50) | 1239 |
| 8100 | New (30) | 1196 |
| 8100 | Competitive (50) | 1005 |
| 8100 | Competitive (30) | 1005 |
| 22500 | New (50) | 6611 |
| 22500 | New (30) | 7011 |
| 22500 | Competitive (50) | 4766 |
| 22500 | Competitive (30) | 4766 |

# I  HYPERPARAMETERS DESCRIPTION AND EXPERIMENTAL SETTINGS

This section outlines the key hyperparameters used in our coarsening and simulation framework for both experiments.

The main hyperparameters used include (excluding GNN architecture' hyperparameters):

- *Learning Rate (lr)*: The learning rate for the Adam optimizer used during training.
- *Physics Loss Weight*: The weight applied to the physics loss in the final loss function.
- *Sigmoid Weight*: Scaling factor for the stability loss (inside sigmoid).
- *Time Step (dt)*: Time step for solving the PDE.
- *Number of epochs*: Defines the number of epochs of algorithm.

## I.1  HYPERPARAMETERS USED IN LOOP SCENARIO

For the loop scenario, the following hyperparameters were used for reduction degree 10%:

- Time Step: 0.0001
- Number of Epochs: 300
- Learning Rate: 0.015
- Sigmoid Weight: 10
- Physics Loss Weight: 15

For reduction degree 7.5%:

- Time Step: 0.0001
- Number of Epochs: 300
- Learning Rate: 0.01
- Sigmoid Weight: 10
- Physics Loss Weight: 20

For reduction degree 5%:

- Time Step: 0.0001
- Number of Epochs: 300
- Learning Rate: 0.01
- Sigmoid Weight: 10
- Physics Loss Weight: 15

## I.2  HYPERPARAMETERS USED IN SINUSOIDAL PERMEABILITY SCENARIO

For reduction degree 10%:

- Time Step: 0.00001
- Number of Epochs: 300
- Learning Rate: 0.015
- Sigmoid Weight: 10
- Physics Loss Weight: 20

For reduction degree 7.5%:

- Time Step: 0.00001
- Number of Epochs: 300
- Learning Rate: 0.01
- Sigmoid Weight: 10
- Physics Loss Weight: 20

For reduction degree 5%:

- Time Step: 0.00001
- Number of Epochs: 300
- Learning Rate: 0.01
- Sigmoid Weight: 10
- Physics Loss Weight: 25

## J    PROXY MODEL SCENARIO

To demonstrate the effectiveness of our approach in predictive tasks at specific measurement points, we carried out experiments that model predictions for future time steps not used during the training stage. In the "loop" scenario with eight sinks (representing critical points), we compared our method against the competing algorithm. Importantly, the training loop is performed only in 1000 timesteps for both algorithms.

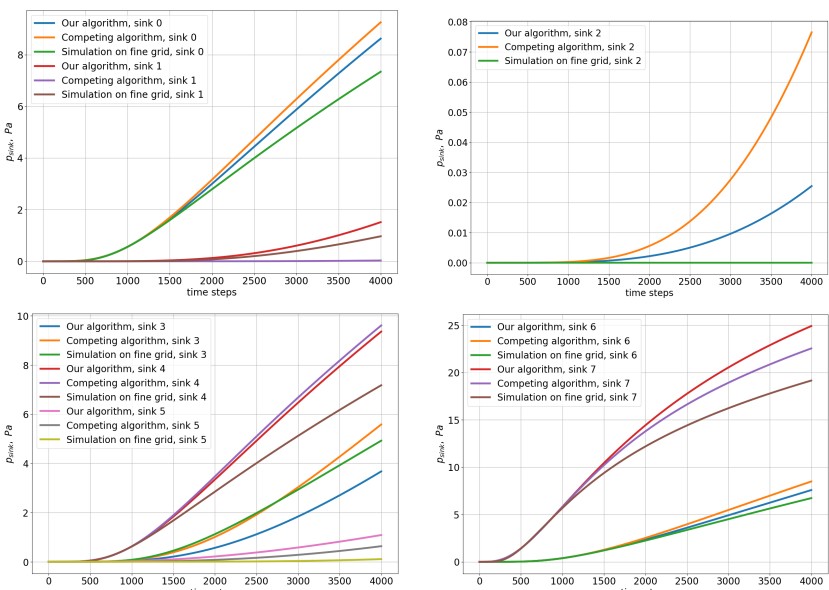

Figure 11: Comparison of $p_s$ obtained by our algorithm and algorithm from Shumilin et al. used for future prediction in *loop scenario*
.

Results show that our algorithm beats the competing algorithm for sinks 0, 1, 2, 4 and 6. Hence, it demonstrates competitive or slightly better performance and, additionally, does not only optimize points but represents a trainable coarsening approach, providing a novel extension to the competing algorithm. Possibly, better choice of hyperparameters could lead to even better results.

Altogether, our algorithm coarsens the grid not using the solution over all the grid but the solution only at several critical points.

# K    COMPARISON RESULTS WITH METIS

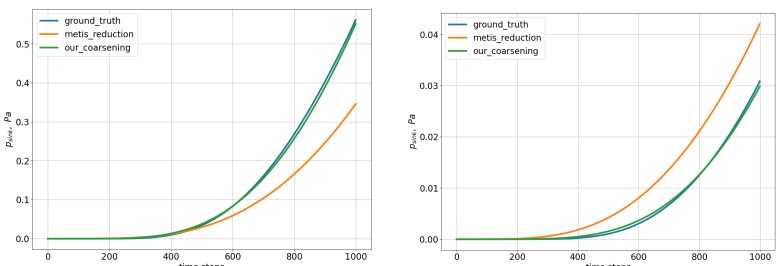

Figure 12: Comparison of $p_s$ obtained by our algorithm and Metis for different permeability scenarios: a) "loop" scenario. b) sinusoidal permeability field.

# L    COMPLEX EXAMPLE RESULTS

In this experiment the grid is initialized with a resolution of $30 \times 30$ points, resulting in a total of 900 grid points. We generate permeability based on a log-uniform distribution while excluding permeability values that are greater than 0.01 and less than 100 (see Appendix G). We apply our algorithm with 10% of the original grid resolution. First, let's demonstrate the results of our algorithm in the zero sink (see Fig. 13).

It can be seen that in this example of the algorithm, more steps are required to converge to a solution, and it is also seen that our loss function has many perturbations. Now let's look at the algorithm from the Shumilin et al. (see Fig. 13). In this case, at one of the steps of the algorithm, the scheme of the explicit solver diverges, which leads to the algorithm stopping. The algorithm we have demonstrated provides stability loss, which prevents the discrepancy of the explicit solution scheme.

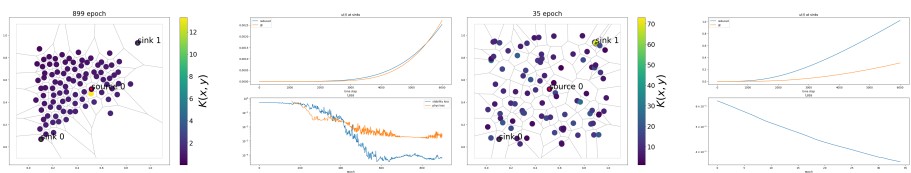

Figure 13: Results of algorithms and pressure measurements in the zero sink: a) our algorithm b) method from Shumilin et al.

We also demonstrate an experiment with less complex permeability field, when we take permeability values from log-uniform distribution and collect values which less than 10 and greater than 0.1. In this case, the algorithm from the Shumilin et al. got a more accurate result, but also broke down at one of the steps (See results in Fig. 14).

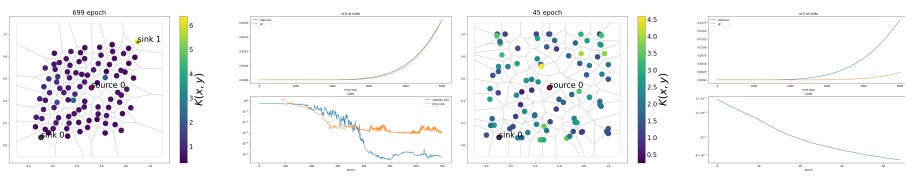

Figure 14: Results of algorithms and pressure measurements in the zero sink: a) our algorithm b) method from Shumilin et al.

# M MOTIVATION FOR IMPLICIT SOLVER

In this appendix, we provide additional details on the implementation of the implicit solver for the diffusion equation and demonstrate its advantages in terms of stability compared to the explicit solver.

The explicit Euler scheme can suffer from stability issues, especially when dealing with large time steps or fine spatial discretizations. Without the inclusion of a stability loss term, the explicit solver may diverge. To address this limitation and broaden the applicability of our method, we implemented an implicit solver, which is known for its superior stability properties.

We conducted a numerical experiment using both the explicit and implicit solvers to solve the diffusion equation with a sinusoidal permeability distribution. In this experiment, we used 100 timesteps, $\tau = 0.01$, Gaussian-like initial condition with no source terms.

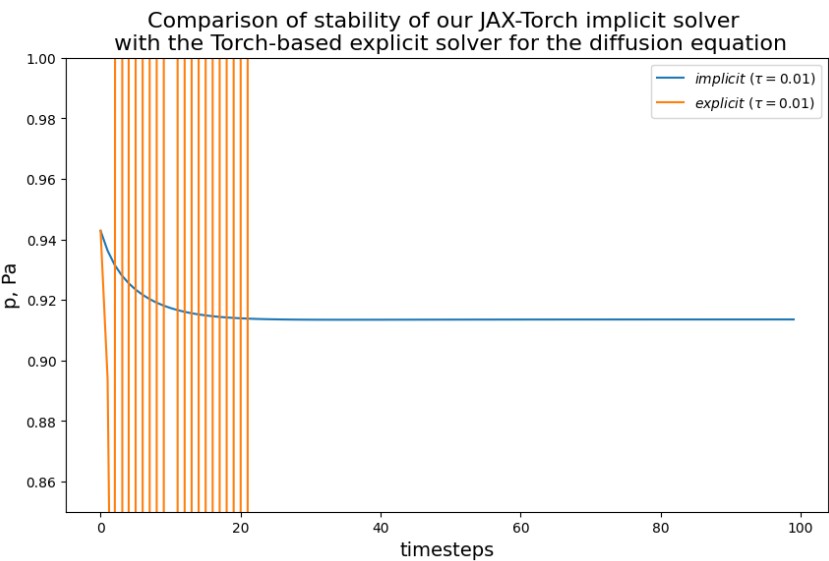

Figure 15: Comparison of pressure distributions obtained from the explicit and implicit solvers for the diffusion equation with sinusoidal permeability. The explicit solver without stability loss diverges, while the implicit solver remains stable.

Figure 15 illustrates the pressure distributions obtained using the explicit and implicit solvers at different time steps. As observed, the explicit solver without the stability loss diverges, leading to non-physical results. In contrast, the implicit solver remains stable and produces accurate pressure distributions throughout the simulation.

By incorporating the implicit solver into our framework, we enhance the stability and extend the applicability of our graph coarsening method to a broader range of physical simulations. The implicit solver serves as a robust building block, particularly in scenarios where the explicit solver may fail due to stability issues. This approach also enables possibility to handle stiff and nonlinear PDEs within our framework, facilitating extensions to more complex problems that require advanced numerical schemes like WENO for nonlinear advection terms.

