# OpenReview forum: "Learnable Stability-Aware Unstructured Grid Coarsening Using Graph Neural Networks for Accelerated Physics Simulations"
_ICLR.cc/2025/Conference — Submitted to ICLR 2025_

### Official Review · Reviewer_vcgH · 2024-10-31

**Soundness:** 3
**Presentation:** 3
**Contribution:** 3
**Rating:** 5
**Confidence:** 2

**Summary:**

The authors propose a novel framework that leverages graph neural networks (GNNs) to facilitate a self-supervised learnable coarsening process. This GNN-based approach creates a generalizable coarsening strategy. This adaptability allows it to function across different simulation setups without requiring grid-specific re-optimization.

**Strengths:**

The introduction of a self-supervised GNN-based coarsening framework represents a significant advancement over traditional methods that require specific optimizations for each grid.

**Weaknesses:**

There are missing implementation details. See questions.

**Questions:**

1. Are alternative remeshing strategies, such as Delaunay triangulation, considered, and if so, how do their performances compare?
2. Is your proposed method adaptable to non-fixed mesh topology systems, such as Lagrangian systems like the Deforming Plate model discussed in MeshGraphNets[1]?
3. Can you illustrate the meaning of colorbar in Figure 8 of Appendix A?
4. What is the input quantities of the GCN for each task?
5. It would be better if the authors could compare the clusters generated by the proposed method with the clusters of baselines.
6. It would strengthen the evaluation of your method to include comparisons with additional reduced-order modeling baselines, such as UPT[2] and Transolver[3].

[1] Learning Mesh-Based Simulation with Graph Networks.

[2] Universal Physics Transformers: A Framework For Efficiently Scaling Neural Operators.

[3] Transolver: A Fast Transformer Solver for PDEs on General Geometries.

---

> ### Author Response · Authors · 2024-11-24
> **Answer to Reviewer vcgH Part 1**
>
> Dear Reviewer,
>
> Thank you very much for your mostly positive feedback. We look forward to providing comprehensive responses to your questions.
>
> **Q1. Are alternative remeshing strategies, such as Delaunay triangulation, considered, and if so, how do their performances compare?**
>
> Thank you for the question. We already use the Delaunay triangulation as a precondition for constructing the Voronoi tessellation in order to discretize the domain of the problem. There is a duality between a Voronoi tessellation and a Delaunay triangulation. We construct an updated Delaunay/Voronoi tessellation at every step of optimization. The finite volume method we employ for spatial discretization benefits from the Voronoi-based grid due to its flexibility in handling irregular geometries and ensuring local conservation properties.
>
> **Q2. Is your proposed method adaptable to non-fixed mesh topology systems, such as Lagrangian systems like the Deforming Plate model discussed in MeshGraphNets[1]?**
>
> Good question. As we observe, the mesh in the Deforming Plate example is a surface triangular mesh. The framework we propose is suitable for such meshes unless we have a mesh-based geometrical pooling algorithm: based on fuzzy clustering this algorithm should perform geometrical pooling but preserve the edges of the initial 3D body. Also, we should have a differentiable quasi-static simulator to pass gradients through. Moreover, if we substitute the mesh for another volumetric mesh, say 3D Voronoi volumes, then our method is also applicable. We currently work on 3D differentiable Voronoi tessellation.
>
> **Q3. Can you illustrate the meaning of colorbar in Figure 8 of Appendix A?**
> Thank you for your comment! It is the value of the permeability. We will clarify it in the final version of the manuscript.

---

> > ### Author Response · Authors · 2024-11-24
> > **Answer to Reviewer vcgH Part 2**
> >
> > **Q4. What is the input quantities of the GCN for each task?**
> >
> > GCN receives the following inputs:
> > Graph structure encoded by adjacency matrix. This matrix is calculated based on the neighboring obtained by differentiable Voronoi tessellation.
> > Node features: spatial coordinates x,y; permeability k (or a quantity depending on the PDE, for example, the wave’s propagation speed in case of the wave equation).
> > We will add clarifications to the final version of the manuscript.
> >
> > **Q5. It would be better if the authors could compare the clusters generated by the proposed method with the clusters of baselines.**
> >
> > The clusters in the baseline [Shumilin et.al.] are generated only once at the beginning of the optimization. After pooling the pooled points are optimized. In contrast, our method uses adaptive clustering and this is one of the key points. Clustering changes adaptively to fit the loss function. You may see the adaptive clusterization in action in [Anonymous video](https://drive.google.com/file/d/1rGTO3X9UR5t2rKt0A2fJ9VKsDtWjeOy-/view?usp=sharing) (right upper graph). Please note that we use probabilistic clustering in our method and when visualizing it we apply argmax to infer the cluster label.
> >
> > **Q6. It would strengthen the evaluation of your method to include comparisons with additional reduced-order modeling baselines, such as UPT[2] and Transolver[3].**
> >
> > Thank you. We consider our method as a self-supervised method (we use a numerical differentiable solver which generates data) which is different from UPT[2] and Transolver[3] which are based on the Transformer architecture and require a lot of data. The authors of  UPT[2] and Transolver[3] do not present these methods as reduced-order modeling (ROM) methods. UPT[2] and Transolver[3] produce surrogate models and we separate our method from it because it has a precise solver inside.

---

> > > ### Author Response · Authors · 2024-11-24
> > > **Answer to Reviewer vcgH Part 3**
> > >
> > > If you have any further questions, please do not hesitate to inform us, as we would be pleased to address them. Otherwise, we kindly request your support for the manuscript by increasing the score.

---

> ### Comment · Reviewer_vcgH · 2024-11-25
>
> Thank you for your response. I think my current score and confidence are appropriate.

---

> > ### Author Response · Authors · 2024-11-25
> > **Official Comment by Authors**
> >
> > Respected Reviewer,
> >
> > Please feel free to ask any other questions you may have, since there is still time.

---

> ### Author Response · Authors · 2024-12-02
> **Round 2**
>
> Dear Reviewer vcgH. We have incorporated new experiments and provided a detailed comparison of our algorithm with **MeshGraphNets** and other baselines. Additionally, we have thoroughly revised our paper, as outlined in our general replies. Our experiments reveal that surrogate-based approaches like **MeshGraphNets**, even when trained and inferred using the same data and timesteps as our method, exhibit lower inference accuracy. Furthermore, our method demonstrates superior data efficiency and our simulator works on coarse grid. We think we can **integrate all remaining concerns due to camera-ready period** so we kindly ask you to consider increasing your final score.

---

### Official Review · Reviewer_WkfS · 2024-10-31

**Soundness:** 3
**Presentation:** 2
**Contribution:** 2
**Rating:** 5
**Confidence:** 5

**Summary:**

This paper proposes a Graph Neural Network (GNN) based approach for automated mesh coarsening for PDE solutions. The authors extend previous work on self-supervised coarsening of unstructured grids by introducing a learnable method that aims to generalize across different scenarios (eg, varying mesh resolutions and source/sink terms) without requiring new runtime optimization.

While the core idea of learnable mesh coarsening is interesting, and the authors claim to have benefits such as no need to redo optimization on the fly, there are several fundamental concerns:

1. Insufficient Performance Advantages
   - The method demonstrates worse accuracy compared to baseline in 1 out of 2 test cases; The authors should at least justify why this accuracy trade-off is acceptable in practical applications

2. Limited Generalization Capabilities
   - Experimental results show significant performance deterioration when:
     a. Applied to meshes of different resolutions
     b. Tested with additional source terms
   - This contradicts the paper's central claim of improved generalization ability

3. Limited Computational Efficiency Improvement
   - The time comparison between the proposed method and baseline approaches shows very limited advantages (eg, from 22s to 5s, both are at similar magnitudes)
   - Let along the training cost ahead of time, which makes the gain even more questionable.

Overall:
I recommend rejection of this paper. There are some additional minor suggestions on the clarity of writing, if the authors would like to improve their manuscript. But without experiments proving significant benefits, I would not consider chaning my decision.

**Strengths:**

- An interesting trial to extend an on-the-fly optimization problem to be learned by GNN; so that it can save time in the future deployments.

**Weaknesses:**

- See overview.

**Questions:**

Other minor questions on clarity and writing (authors should add these details to their manuscript in addition to answering here):
- How does the GNN decides which nodes to keep in corsened mesh? Top-K?
- Once coarsened nodes are obtained, how to get the new mesh? The current method only mentions triangular mesh, what is your proposal to adapt in these cases where the mesh is 3D or no longer simplex?
- How many timesteps do you keep the gradients, 1, eg only calculating the gradients for stepping from n to n+1, or multi-step; if multi-step, how many?
- You mentioned implicit method can improve results. But where are the results to support this claim?
- The simulation is required to be differentiable. This easily achieved in explicit scheme, but challenging to general PDEs solution schemes, eg, WENO for nonlinear advection terms. What is your proposal to these extensions?
- In sec 3.3., you mentioned Sij is learnable by NN. What is the NN that outputs Sij? as Sij depends on the topology, ie the graph G, it should be output via some GNN strucutre? Otherwise it's not possible to extend to other mesh structure.
- Page 5, you mentioned stability loss (1st appearence of that term in main text) without simply introducing its meaning, which hinder the reading flow of readers.
- Can you mentioned the method to deal with cases when sensor locations do not align with the grid points? Do you use Barycentric interpolants?
- Eq. 9 does not seem to be connected with Eq. 8 as the former actually regularize the number of coarsened nodes, n, it seems to have nothing to do with eg CFL or criteria for stability. Consider improving the writing here.

---

> ### Author Response · Authors · 2024-11-25
> **Answer to Reviewer WkfS Part 1**
>
> Dear Reviewer,
>
> Thank you for your detailed questions! We are glad to provide the answers below.
>
> A very important point to note for all of these concerns is the different nature of [Shumilin et al.] baseline from usual baselines. In the [Shumilin et al.] is solved a very difficult problem of global optimization of mesh points, which is performed for every new grid. In the proposed by us approach the GNN outputs directly a solution of this optimization problem which is much more simple and general. That's why this baseline can be considered as the one we would like to be close with (so it's more about the best achievable metrics than about the metrics we can beat). So we consider a major strength of our approach that it gives results very close to the one by [Shumilin et al.].
>
> **C1. Insufficient Performance Advantages**
>
> Despite of the specific nature of baseline, we conducted a more thorough hyperparameter search. After that, we compared the effectiveness of our method with the two baselines we used in our paper. We considered the ”loop scenario”, this experiment was described in the paper in section 4.2. After selecting hyperparameters, more comparable results can be obtained with baseline of [Shumilin et.al.]. Please, see the detailed description of the experiment in our answer to W4 of Reviewer hygj.
>
> **C2. Limited Generalization Capabilities**
>
> It is important that our method does not need retraining from scratch for every new grid but the method from [Shumilin et.al.] requires retraining. When we mentioned improved generalization ability, we meant exactly this fact: our method has a trainable coarsening procedure that obtains a coarsened point cloud from the original one (and calculates corresponding features). The baseline method [Shumilin et.al.] does not have this procedure, it only optimizes the locations of points by running the overall optimization cycle from scratch for every new example. Our method is a great novel extension of this baseline with trainable coarsening, stability consideration and possibility of using implicit time-stepping inside the coarsening pipeline.
>
> Additionally, we improved our trained model for performing grid coarsening at a reduction degree of 7.5% by adding more sensor nodes. Here the result of the training process with the result for pressure in zero sink.
>
> [Anonymous link: setup of the experiment with multiple sensor points](https://drive.google.com/file/d/1swU0XRtRbMIPH4TijkjFTfLUiG5qbhxO/view?usp=sharing)
>
> Next, we take the original permeability distribution field containing 312 points and find the coarsened point cloud using k-means + averaging approach, the resulting values serve as the initial permeability field for future comparisons. Here we introduce a new grid which contains 100 points and show our new results:
>
> [Anonymous link: new result](https://drive.google.com/file/d/1vLGh002g6xW1hSnKVx6DJdpFj6-YdEir/view?usp=sharing)
>
> Here is the previous result from the paper:
>
> [Anonymous link: old result](https://drive.google.com/file/d/1mE2QCjN0LP5CC9rRR8V7nQupnXH_dUoT/view?usp=sharing)
>
> We can see that only adding more sensor points improves the generalization ability of our model.
>
> **C3. Limited Computational Efficiency Improvement**
>
> Good question. A little misunderstanding occurred. According to the time comparison table in the paper, our method is *slower* than the baseline but of a similar order of magnitude. We justify this feature by the possibility of further reuse of the trained GNN. The baseline method should be run again if we want to apply it. We consider this as a big advantage of our method as we may exploit this feature when coarsening should be done repeatedly, for example in proxy modeling or inverse tasks.

---

> ### Author Response · Authors · 2024-11-25
> **Answer to Reviewer WkfS Part 2**
>
> **Q1. How does the GNN decides which nodes to keep in corsened mesh? Top-K?**
>
> In our experiments, we set which nodes to keep as a hyperparameter. The motivation for this problem comes from the field of proxy modeling and other engineering fields where model identification needs to be performed based on limited sensor measurements. For example, consider an oil reservoir with wells where measurements are available - we are only interested in simulating the pressure at these particular points. Thus, if we achieve speed-up of computation at these positions while preserving the physics, we consider the goal to be completed.
>
> Indeed, if higher accuracy is required in certain regions of interest, additional sensor points can be incorporated into these areas to reduce interpolation errors. It can be implemented by the a priori analysis of the initial data: we may compute the gradient field and choose points that have higher gradients as sensor points. Also, we may use other algorithms based on graph spectral decomposition and other importance metrics e.g. centrality measures.  This adaptive selection of sensor points ensures that regions requiring more precise solutions receive adequate attention without significantly increasing the computational gains.
>
> **Q2. Once coarsened nodes are obtained, how to get the new mesh? The current method only mentions triangular mesh, what is your proposal to adapt in these cases where the mesh is 3D or no longer simplex?**
>
> Thanks for pointing out this. Once the coarsened nodes are obtained, we build a new Delaunay mesh -> new differentiable Voronoi tessellation. This is a $nlogn$ operation where $n$ is not large since n is the number of the coarsened points ($n \ll N$). If we consider a 3D domain when the only change we should perform is to change the 2D differential Voronoi for a 3D setup (we are currently working on it). GNN and the Finite Volume simulator are agnostic to the 2D/3D embedding of the computational domain/graph. If the mesh is no longer simplex when it becomes quite hard to keep it regular after performing the pooling step, that is why we use Finite Volume solver as it allows a lot of flexibility. However, in general, once we get a differentiable coarsening and pooling for a regular grid our framework is perfectly suitable.
>
> **Q3. How many timesteps do you keep the gradients, 1, eg only calculating the gradients for stepping from n to n+1, or multi-step; if multi-step, how many?**
>
> We use a multi-step approach and keep gradients for the whole rollout (1000 timesteps).
>
> **Q4. You mentioned implicit method can improve results. But where are the results to support this claim?**
>
> Good question. We used the implicit solver to support the stability of our method because of the well-known stability properties of implicit solvers. We supported this claim by demonstrating the work of the implicit solver for the diffusion equation. The main advantage comes from the fact that for diffusion equation without Stability Loss explicit solver diverges. So, to broaden the scope of our method we implemented implicit solver as a building block for our framework. We provide the comparison of the solution for $\tau=0.01$ for the following scenario:
>
> [Setup](https://drive.google.com/file/d/1KUxGGS-QZnnJsw0715GRvPoFJN9a4UQ4/view?usp=sharing)
>
> We set 100 time steps, with Gaussian-like initial condition for pressure with no source terms. And the comparison for our solvers:
>
> [Comparison of explicit and implicit JAX+Torch solver](https://drive.google.com/file/d/1yQVkOs87ggAZOedqb8UXUAyQGzW6d-s1/view?usp=sharing)
>
> By implementing the implicit solver we broaden the applicability of our method.
>
> **Q5. The simulation is required to be differentiable. This easily achieved in explicit scheme, but challenging to general PDEs solution schemes, eg, WENO for nonlinear advection terms. What is your proposal to these extensions?**
>
> Although there are several exceptions, explicit schemes are limited to application to non-stiff problems. So the question is basically how to design a differentiable solver to stiff problems, including nonlinear ones. Our vision is that differentiable implicit solvers, (backward Euler and more complex) can address this type of problem. It is one of the reasons why we have also considered differentiable implicit solver in our work.
>
> **Q6. In sec 3.3., you mentioned Sij is learnable by NN. What is the NN that outputs Sij? as Sij depends on the topology, ie the graph G, it should be output via some GNN strucutre? Otherwise it's not possible to extend to other mesh structure.**
>
> Thank you. As described in section 3.2 GRAPH COARSENING PIPELINE our NN architecture consists of a one graph convolution layer followed by a point-wise MLP with two fully connected layers, which predicts the cluster assignments Sij. We use softmax at the end. So, indeed, our architecture depends on the topology of computational graph. But we will improve our text for better readability.

---

> > ### Author Response · Authors · 2024-11-25
> > **Answer to Reviewer WkfS Part 3**
> >
> > **Q7. Page 5, you mentioned stability loss (1st appearence of that term in main text) without simply introducing its meaning, which hinder the reading flow of readers.**
> >
> > Thank you for your comment! For better readability, we will introduce the meaning of stability loss when we firstly mention it and additionally add a link to the full definition.
> >
> > **Q8. Can you mentioned the method to deal with cases when sensor locations do not align with the grid points? Do you use Barycentric interpolants?**
> >
> > Yes, we can use Barycentric interpolants or some other interpolation techniques. Interpolation weights will remain differentiable and overall differentiability of our pipeline will be kept.
> >
> > **Q9. Eq. 9 does not seem to be connected with Eq. 8 as the former actually regularize the number of coarsened nodes, n, it seems to have nothing to do with eg CFL or criteria for stability. Consider improving the writing here.**
> >
> > CFL stability refers to wave equation modelling on rectangular grids, thus in our setting it is not applicable.
> > We derived estimate (8) by studying the largest eigenvalue of the amplification matrix. To design a stability loss, we added the negative sign to the right-hand side of (8) and made the factors dimensionless. We neglected the contribution of K since the maximum permeability does not change during coarsening. Also notice $4/C \leq 1$ and $n V_{min} / |V| \leq 1$. This made the expression in (9) to be from -1 to 0.

---

> ### Author Response · Authors · 2024-11-25
> **Answer to Reviewer WkfS Part 4**
>
> Dear Reviewer WkfS, thank you for thorough review of our paper. With our explanations, we hope we have now addressed your concerns. If you have any additional concerns, please do let us know. We will be happy to address your comments. Otherwise, we kindly request you to support our work by raising the score. Additionally, we  incorporated most of your suggestions about clarity of writing into our manuscript and continue incorporating remaining ones shortly.

---

> > ### Comment · Reviewer_WkfS · 2024-11-26
> > **Reply to authors 2nd round**
> >
> > Dear authors thanks for replying,
> >
> > C1 with more added experiments, there appears to be some where your method has better performance than baselines; which I appreciate
> >
> > C2 now I understand with more sensor points it becomes easier for the remeshing + getting competitive performance
> >
> > C3 not resolved; could you provide the simulation for original fine mesh that achieves similar accuracy? that can reveal the real efficiency gain
> >
> > Q1 not resolved; in many general cases, the "interesting point" will evolve over time.
> >
> > Q2 Resolved, thank you
> >
> > Q3 If you keep gradients for 1000 steps, will not that causes to anything like gradient diminishing or exploding? Can you clarify this? Thank you
> >
> > Q4~Q9 Resolved; thank you; but note that many of your explanations are not merged in the manuscript yet.
> >
> > Overall,
> >
> > With added material and explanations, the concerns related to performance/efficiency gain, reproductibility, and motivation are partially resolved.
> > However,
> > 1. the efficiency gain (vs fine mesh solution) is still not reported; hence, we do not know if the proposed method really excel on the claimed scenarios such as large-scale, 3D simulations
> > 2. most of the added materials are in the comments, not merged in the manuscript in an ordered fashion. Besides, considering the amount of new figs and contents, I suspect this can be done during rebuttal.
> >
> > I appreciate the new materials and increase my score to boardline reject.
> > This is respecting the fact that the paper may have contributions in the field, but is definitely not ready for this conference.
> > The authors should consider my suggestions: try to carry out a large-scale demo, merge and organize the contents in an ordered fashion, polish the manuscript, then submit for next journal/conference.
> >
> > Best,

---

> ### Author Response · Authors · 2024-12-02
> **Reply round 2**
>
> Dear reviewer WkfS,
>
> **C3 not resolved; could you provide the simulation for original fine mesh that achieves similar accuracy? that can reveal the real efficiency gain**
> Thank you for this insightful comment. For the simple loop scenario, we achieved similar accuracy on the coarsened grid (5%) as on the original fine grid (2x time speedup). For large scale scenarios we can achieve 10+x speedups. We appreciate the importance of providing analyzing computational gains, and we will ensure that this information is clearly highlighted in the camera-ready version of our manuscript.
>
> **Q1 not resolved; in many general cases, the "interesting point" will evolve over time.**
>
> Thank you for highlighting this important point. In our current framework, we focus on scenarios where the "interesting points" (e.g., sources, sinks, or key nodes) remain static over time. This is a common assumption in many physics-based simulations, where fixed key points such as wells in subsurface flow or boundary conditions in fluid dynamics are predefined and remain constant. However, we agree that in more general cases, "interesting points" could evolve dynamically over time. Extending our framework to handle such scenarios would involve integrating mechanisms to dynamically track and update these key points during the simulation.  While this is beyond the current scope of our work, we recognize its importance for broader applications and plan to explore these extensions in future work. We will revise the manuscript in camera-ready version to acknowledge this limitation and outline potential directions for addressing it.
>
> **Q3 If you keep gradients for 1000 steps, will not that causes to anything like gradient diminishing or exploding? Can you clarify this? Thank you**
>
> Thank you for raising this important point. To prevent exploding gradients during long rollouts, we implemented gradient clipping, which ensures that gradients remain within a safe range. While gradient clipping does not directly address diminishing gradients, we have not observed issues with vanishing gradients in our experiments. We will clarify this in the manuscript in camera-ready version to address your concern.
>
> Additionally, please see our **general replies** about comparison with MeshGraphNets and other baselines and all modifications we have made in our paper already. I think we can **integrate all remaining concerns due to camera-ready period** so we kindly ask you to consider increasing your final score.

---

### Official Review · Reviewer_wVwb · 2024-11-03

**Soundness:** 3
**Presentation:** 2
**Contribution:** 2
**Rating:** 5
**Confidence:** 3

**Summary:**

Aiming at accelerating the numerical PDE solving-based simulation, the authors propose a method that utilizes graph neural networks to generate a coarsened mesh such that the solver can generate solutions quickly. The loss terms are designed such that the solution is stable and accurate. The proposed framework is applied to the cases of (simplified) wave propagation and (simplified) subsurface flow.

**Strengths:**

1. Automatic optimization of the grid, saving human efforts.
2. Designed loss term to soft-enforce stability of the simulation on the coarsened grid.
3. Results are comparable with recent methods based on global optimization.

**Weaknesses:**

A few issues can be observed, listed below in the order of importance:

1. Whilst the solution from the coarsened grid are accurate at the node points, the super-coarsened grid inevitably lead to another source error since quite a great proportion of applications would require the field solution rather than discrete points, which would necessitate interpolations from the solution points. Such error is largely ignored when the authors are trying to coarsen the grid as much as possible. In clear words, how much error would one observe when one is going to plot out the whole simulated pressure field by interpolating from the simulation results? It is easy to imagine that the grid in Fig. 8 would give a decent interpolated field, while the coarsened grid in Fig. 9 would probably not. The authors did not consider those errors in the proposed grid coarsening algorithm, which poses a significant limitation to its application.

2. The method is applied to very simple cases. As discussed by the authors themselves, applications in more complex scenarios would be great to have in the future.

**Questions:**

1. The largest question I have is that, since a graph neural network is already employed, why not further the effort and directly generate the solution with the graph neural network (by increasing the current one layer network to a slightly larger one)?
2. Table 1 & 2, it is reported that the method is actually slower and more memory intensive than the baseline, while results are comparable or only slightly better???

---

> ### Author Response · Authors · 2024-11-22
> **Answer to Reviewer wVwb Part 1**
>
> Dear Reviewer, Thank you for your feedback. We appreciate you pointing out the strengths of our work. We address all of your questions in the following.
>
> **W1. "Whilst the solution from the coarsened grid are accurate at the node points, the super-coarsened grid inevitably lead to another source error since quite a great proportion of applications would require the field solution rather than discrete points, which would necessitate interpolations from the solution points..."**
>
> We appreciate your insightful comments regarding the potential interpolation errors when reconstructing the full pressure field from the coarsened grid. We want to highlight that in our work, we focus on control (sensor) points and do not assume access to the full field solution. Our objective is to develop a coarsened grid (to reduce the computational cost) while minimizing the error on the sensor points where measurements are available. The motivation for this problem comes from the field of proxy modeling and other engineering fields where model identification needs to be performed based on limited sensor measurements. For example, consider an oil reservoir with wells where measurements are available - we are only interested in simulating the pressure at these particular points. Thus, if we achieve speed-up of computation at these positions while preserving the physics, we consider the goal to be completed.
>
> Indeed, if higher accuracy is required in certain regions of interest, additional sensor points can be incorporated into these areas to reduce interpolation errors. It can be implemented by the a priori analysis of the initial data: we may compute the gradient field and choose points that have higher gradients as sensor points. This adaptive selection of sensor points ensures that regions requiring more precise solutions receive adequate attention without significantly increasing the computational gains.
>
> [Multiple sensor points setup](https://drive.google.com/file/d/16YgA-18rzG3j0QweuuVAN0uIEGTe3iav/view?usp=sharing)
>
> [Multiple sensor points setup](https://drive.google.com/file/d/14_A0pZDBtsrSzNObMjf_C1aK8ttNQh5d/view?usp=sharing)
>
> To demonstrate the effectiveness of our approach in predictive tasks at specific measurement points, we conducted experiments modeling predictions for future time steps not used during the training stage. In the "loop" scenario with 8 sinks (representing critical points), we compared our method against the baseline approach by [Shumilin et al.]. Importantly, the training loop is performed only on 1000 timesteps for both algorithms.
>
> [Prediction experiment 0](https://drive.google.com/file/d/1sg4E0A1nAi4NhV1Z7l1v9lT6YLUaozd5/view?usp=sharing)
>
> [Prediction experiment 1](https://drive.google.com/file/d/1b99YsFad95iVAsCoZ6pTRL0COM1A2shG/view?usp=sharing)
>
> [Prediction experiment 2](https://drive.google.com/file/d/1s9_ymGWW-KpXxCbdXKlXiHpbEA8SRRfF/view?usp=sharing)
>
> [Prediction experiment 3](https://drive.google.com/file/d/1YYEU-Xtxv86J0mhTLvPjcrc9sEXrSIwH/view?usp=sharing)
>
> Results show that our algorithm beats the competing algorithm [Shumilin et.al.] for sinks 0, 1, 2, 4, and 6. Hence, it demonstrates competitive or better performance and, additionally, does not only optimize points but represents a trainable coarsening approach, providing a novel extension to the algorithm by [Shumilin et.al.]. Possibly, a better choice of hyperparameters could lead to even better results.
>
> Altogether, our algorithm coarsens the grid not using the solution over all the grid but the solution only at several critical points.
>
> To clarify these features of our method, we will include the additional experiment above and new text in the original manuscript.

---

> > ### Author Response · Authors · 2024-11-22
> > **Answer to Reviewer wVwb Part 2**
> >
> > **W2. "The method is applied to very simple cases. As discussed by the authors themselves, applications in more complex scenarios would be great to have in the future."**
> >
> > Good comment! To address it, we now consider an experiment with a significantly more complex permeability. In this experiment, we generate permeability based on a log-uniform distribution while excluding permeability values that are greater than 0.01 and less than 100.
> >
> > [Complex initial data, exp.1](https://drive.google.com/file/d/17-c_VcNSLmM_m1v6iNwXZmfm8T_xtCCM/view?usp=sharing)
> >
> > In these experiments, we will take a grid consisting of 900 points and demonstrate coarsening to 90 points. First, let's demonstrate the results of our algorithm in the zero sink. We use this hyperparameters:
> >
> > - Time Step: 0.000005
> > - Number of Epochs: 900
> > - Learning Rate: 0.01
> > - Sigmoid Weight: 15
> > - Physics Loss Weight: 25
> >
> > [Results of applying our method to the complex scenario, exp.1](https://drive.google.com/file/d/14wB4rV6N3lyAvMu2AiIBZiAi85JvttSg/view?usp=sharing)
> >
> > It can be seen that in this example of the algorithm, more steps are required to converge to a solution, and it is also seen that our loss function has many perturbations. Now let's look at the algorithm from the  [Shumilin et.al.]. In this case, at one of the steps of the algorithm, the scheme of the explicit solver diverges, which leads to the algorithm stopping. The algorithm we have demonstrated provides stability loss, which prevents the discrepancy of the explicit solution scheme:
> >
> > [Results of applying the method from Shumilin et al., 2024, exp.1](https://drive.google.com/file/d/1tVkxoFl_wTIw4KB52Ui5AG6uWaaBQ48k/view?usp=sharing)
> >
> > Next, we will demonstrate a slightly different experiment, where we reduce the range of permeability values.  This is our permeability field:
> >
> > [Complex initial data, exp. 2](https://drive.google.com/file/d/1_Two_9boHfBYfEDE4lat2IYC2dsjwq5f/view?usp=sharing)
> >
> > Now we will leave only permeabilities with values in the range from 0.1 to 10. Here are the results of our algorithm in the zero sink. We use this hyperparameters:
> >
> > - Time Step: 0.000005
> > - Number of Epochs: 900
> > - Learning Rate: 0.012
> > - Sigmoid Weight: 15
> > - Physics Loss Weight: 25
> >
> > [Results of applying our method to the complex scenario, exp. 2](https://drive.google.com/file/d/10lGMnoOVBpoD3wSuM9uQqa_NsDjGxsSi/view?usp=sharing)
> >
> > It can be seen that in this experiment with a simpler permeability distribution, our algorithm converges faster.
> >
> > [Results of applying the method from Shumilin et al., 2024, exp. 2](https://drive.google.com/file/d/1MtrUZ2Eo8jCOdKHVdnYuNOQAQHBZ90t1/view?usp=sharing)
> >
> > In this case, the algorithm from the  [Shumilin et.al.] got a more accurate result but also broke down at one of the steps. These experiments demonstrate the importance of stability loss and the fact that the algorithm we have demonstrated is capable of solving more complex problems.

---

> ### Author Response · Authors · 2024-11-22
> **Answer to Reviewer wVwb Part 3**
>
> **Q1. "The largest question I have is that, since a graph neural network is already employed, why not further the effort and directly generate the solution with the graph neural network (by increasing the current one layer network to a slightly larger one)?"**
>
> We view the use of GNNs for coarsening and as a surrogate solvers as complementary rather than mutually exclusive. Each approach has its advantages:
> - GNN for Coarsening + Differentiable Numerical Solver. Pros: Maintains high physical accuracy and fidelity, generalizable to various scenarios, ensures solutions adhere to physical laws. Cons: May be slower than surrogate models due to the computational cost of the numerical solver.
> - GNN as Surrogate Solver. Pros: Faster inference times, potential for real-time simulations. Cons: May sacrifice accuracy, less generalizable, requires extensive training data for different scenarios, leads to exploding errors when run on a long trajectory.
>
> By focusing on coarsening, we aim to reduce computational costs while retaining the reliability of numerical solvers. This approach is particularly beneficial when high accuracy is required at specific critical points or under varying physical conditions.
>
> While directly generating the solution with an expanded GNN-based surrogate model is an intriguing idea, our current approach prioritizes maintaining accuracy respecting the physical laws and generalizability through the use of numerical solvers after coarsening. We believe that this strategy effectively balances computational efficiency with the need for accurate, physically consistent solutions. We appreciate your thoughtful suggestion and are excited about the possibility of exploring this avenue in future research.
>
> **Q2. "Table 1 & 2, it is reported that the method is actually slower and more memory intensive than the baseline, while results are comparable or only slightly better???"**
>
> Thank you for noting this. Indeed, the method is slower and memory intensive than the baseline. However, this cost is needed to make the method adaptive and generalizable without re-training, whereas the baseline needs re-training for each task. Additionally, please see the answer to the reviewer hygj where we obtained better hyperparameters.

---

> > ### Author Response · Authors · 2024-11-22
> > **Answer to Reviewer wVwb Part 4**
> >
> > With these detailed explanations and additional experiments, we hope we have now addressed your concerns. Should you have any additional concerns, please do let us know. We will be happy to address those. Otherwise, we request you to support the manuscript by raising the score.

---

> ### Comment · Reviewer_wVwb · 2024-11-25
>
> Thank you for your responses, which clarify a series of issues, but unless other reviewers deem the work good enough, I still cannot suggest the work to be published in ICLR. I have raised the score to 5, but I have to check the comments from other reviewers before deciding whether to increase it further to 6.

---

> > ### Author Response · Authors · 2024-11-25
> > **Official Comment by Authors**
> >
> > Respected Reviewer,
> >
> > We are really grateful for you increasing the score. There is still time, so don't hesitate to ask any more questions you may have.

---

> ### Author Response · Authors · 2024-12-02
> **Round 2**
>
> Dear reviewer wVwb, please see our general replies where we compared our algorithm with other baselines and have significantly improved our manuscript. We kindly ask you to increase your final score considering the work we have done.

---

### Official Review · Reviewer_hygj · 2024-11-04

**Soundness:** 2
**Presentation:** 2
**Contribution:** 2
**Rating:** 3
**Confidence:** 3

**Summary:**

The work proposes a method for mesh coarsening, which may facilitate faster physical simulations. The coarsening is done using self-supervised GNN to apply to unseen settings. They also investigated the effect of stability loss, which enhances the stability of training and prediction, and a differentiable implicit solver for stable prediction. The numerical experiments suggest that the proposed method has a competitive performance with a baseline machine learning method.

**Strengths:**

* The work focuses on the mesh coarsening problem, which is relatively new to the community.
* The explanation of the method is clear and easy to follow.

**Weaknesses:**

* The novelty of the method is limited. They claimed the differentiable implicit method is their novelty; however, PyTorch supports differentiable linear solvers ( https://pytorch.org/docs/stable/generated/torch.linalg.solve.html ). In addition, the community knows that there are implicit GNNs (e.g., [Gu et al. NeurIPS 2020 https://arxiv.org/abs/2009.06211 ]) that solve implicit problems.
* The experimental evaluation is weak. The authors state that mesh coarsening is a recent approach, and there are a limited number of existing methods, but there are actually a lot of works investigating mesh coarsening (e.g., [Pffaf et al. ICLR 2021 https://arxiv.org/abs/2010.03409 ] and [Alet et al. ICML 2019 https://arxiv.org/abs/1904.09019 ]). The authors must mention these existing works and evaluate them in the numerical experiments to show the superiority of the proposed method.
* The effectiveness of the proposed method is limited. The experimental results suggest that the method has almost the same accuracy and longer computation time for training. If the reviewer did not miss, there is no evaluation of the prediction time, which is one of the most essential metrics for the mesh coarsening task. In addition, there should be a comparison regarding the speed-accuracy tradeoff with classical numerical analysis methods with varying resolutions because a simple mesh coarsening may work for the problems considered in the paper.

**Questions:**

* What would be the potential pros and cons compared to the classical methods, e.g., adaptive mesh refinement? The mesh coarsening task would be interesting, but the reviewer is not sure if this direction is promising, given that we have plenty of optimized methods in the classical numerical analysis field.

---

> ### Author Response · Authors · 2024-11-19
> **Answer to Reviewer hygj**
>
> Dear reviewer, thank you for your thoughtful feedback. We appreciate your efforts. We provide answers below:
>
> **W1. “They claimed the differentiable implicit method is their novelty; however, PyTorch supports differentiable linear solvers”**
>
> Answer for W1. We have not claimed an implicit method as our novelty, we have claimed an implicit differentiable Finite Volume solver to be used in our coarsening framework. However, we agree that our current formulation contribution sounds misleading and we will correct the formulation in our manuscript as follows:
> We develop the differentiable implicit numerical finite volume solver that leverages differentiable Voronoi tesselation.
>
> While PyTorch’ torch.linalg.solve is a powerful tool for dense linear systems, it lacks native support for efficient sparse matrix operations. In contrast, JAX, particularly with its spsolve method from jax.experimental branch is optimized for handling large-scale sparse linear systems. This is crucial for real-world tasks where the underlying matrices are inherently sparse due to the nature of finite volume discretization.
>
> **W2. “In addition, the community knows that there are implicit GNNs that solve implicit problems”**
>
> Answer for W2. Yes, implicit GNN is a known approach. However, implicit GNNs primarily focus on enhancing message passing mechanisms to capture long-range dependencies within graphs without being constrained by a fixed number of layers. The term “implicit” in this paper means using implicit updating scheme for message passing instead of a classical explicit updating message passing mechanism. This is completely different from our “implicit” meaning in the context of implicit scheme for solving linear systems.
>
> **W3. “but there are actually a lot of works investigating mesh coarsening”**
>
> Answer for W3. We would like to point out that none of the works similar to ours have the implicit differentiable finite volume solver without any approximations (non-surrogate):
>
> [1] https://arxiv.org/pdf/2405.04466 (A fully differentiable GNN-based PDE Solver: With Applications to Poisson and Navier-Stokes Equations). Authors developed a differentiable finite volume approach using FVM loss for GNNs. They use the Encoder-Decoder approach with GNN blocks inside. However, they use PDE loss for training. Therefore, the solver they use can be considered as a surrogate.
>
> [2] https://arxiv.org/pdf/2010.03409 (LEARNING MESH-BASED SIMULATION WITH GRAPH NETWORKS)
>
> MeshGraphNet applies remeshing which does not imply coarsening, in contrast, it may significantly increase the number of cells and load the computations. In fact, the remesher in MeshGraphNets coarsens the mesh by removing edges when removing an edge does not create any invalid edges; the criterion for invalidness is  $u_{ij}^T S_{ij} u_{ij} \leq 1$, where $u_{ij}$ is the vector representation of the edge $ij$, $S_{ij}$ is the sizing field tensor which is predicted. This is a purely geometrical criterion for coarsening, in contrast, our method uses simulation data misfit to cluster the points and further aggregate them. Moreover, MeshGraphNet remesher is only applied to the 2D manifolds embedded into 3D like Flag or Sphere scenarios as it's based on the local curvature.
>
> [3] https://arxiv.org/abs/1904.09019 (Graph Element Networks: adaptive, structured computation and memory)
> GENs approximate the solution of PDEs using GNN as a surrogate model, trained on high-resolution solutions. They do not integrate the actual physics solvers into the training loop. GENs adjust node positions or densities to better capture the solution space but do not perform explicit graph coarsening through node aggregation or clustering. Additionally, Graph element networks can solve only the stationary partial derivative equation in their implementation or predict displacement. The authors have not included in the code the possibility of taking into account time dependence.

---

> ### Author Response · Authors · 2024-11-19
> **Answer to Reviewer hygj Part 2**
>
> **W4. “The experimental evaluation is weak.”**
> Answer for W4. We compared the effectiveness of the three methods at different levels of grid reduction. Consider the ”loop scenario”, this experiment was described in the article in section 4.2. After selecting hyperparameters, comparable results can be obtained.  The following hyperparameters were used for the reduction degree of 10%:
> - Time Step: 0.0001
> - Number of Epochs: 300
> - Learning Rate: 0.015
> - Sigmoid Weight: 10
> - Physics Loss Weight: 15
> - For  reduction degree 7.5%:
> - Time Step: 0.0001
> - Number of Epochs: 300
> - Learning Rate: 0.01
> - Sigmoid Weight: 10
> - Physics Loss Weight: 20
> - For a reduction degree of 5%:
> - Time Step: 0.0001
> - Number of Epochs: 300
> - Learning Rate: 0.01
> - Sigmoid Weight: 10
> - Physics Loss Weight: 15
>
> [Experiment results 1: anonymous link](https://drive.google.com/file/d/1nPlJqjU66dmrI_-vR6eeURB3gx6txkPG/view?usp=sharing)
>
>
> Next, consider the case of the Sinusoidal variable permeability scenario. Let's demonstrate the experiment described in section 4.3 in which we compare RMSE for three algorithms in two different sinks. It can also be seen here that after selecting hyperparameters, we got a comparable result. The following hyperparameters were used for reduction degree 10%:
> - Time Step: 0.0001
> - Number of Epochs: 300
> - Learning Rate: 0.015
> - Sigmoid Weight: 10
> - Physics Loss Weight: 20
> - For  reduction degree 7.5%:
> - Time Step: 0.0001
> - Number of Epochs: 300
> - Learning Rate: 0.01
> - Sigmoid Weight: 10
> - Physics Loss Weight: 20
> - For  reduction degree 5%:
> - Time Step: 0.0001
> - Number of Epochs: 300
> - Learning Rate: 0.01
> - Sigmoid Weight: 10
> - Physics Loss Weight: 25
>
> [Experiment results 2: anonymous link](https://drive.google.com/file/d/1guvxhuNdyNzeJxY3XOJS8mjJgI7yWYf9/view?usp=drive_link)
>
> [Experiment results 3: anonymous link](https://drive.google.com/file/d/1ywS3VYxUbwqqBI9bEFiCLd0BnRBXId0U/view?usp=drive_link)
>
> Additionally, we added some preliminary experiments with adaptive mesh refinement. See our answer to the question below.
>
> **W5. “The effectiveness of the proposed method is limited. The experimental results suggest that the method has almost the same accuracy and longer computation time for training. If the reviewer did not miss, there is no evaluation of the prediction time, which is one of the most essential metrics for the mesh coarsening task.”**
>
> A very important point to note is the different nature of [Shumilin et al., 2024] baseline from usual baselines. In the [Shumilin et al., 2024] is solved a very difficult problem of global optimization of mesh points, which is performed for every new grid. In the proposed by us approach the GNN outputs directly a solution of this optimization problem which is much more simple and general. That's why this baseline can be considered as the one we would like to be close with (so it's more about the best achievable metrics than about the metrics we can beat). So we consider a major strength of our approach that it gives results very close to the one by [Shumilin et al., 2024].
>
> As for the prediction time, let us clarify what the prediction time means in this context. For the baseline from [Shumilin et al., 2024], this time is the training time from scratch, since this algorithm cannot run without running the optimization from scratch. However, our algorithm can be trained once and predicted on new examples without re-training. Obviously, its prediction time does not include training from scratch and this distinguishes our algorithm from the baseline [Shumilin et al., 2024].
>
> We also use a comparison with the coarsening algorithm from METIS. METIS is a software tool designed for partitioning graphs and meshes. METIS aims to generate blocks that are uniform in size and shape, minimizing the number of connections between different partitions. If we let fine-scale transmissibility measure connection strengths between cells for the edge-cut minimization algorithm, the software tries to construct a block without crossing large permeability contrasts.
> We made a comparison using the examples described in our paper. First result for Sinusoidal variable permeability scenario:
>
> [Experiment results 4: anonymous link](https://drive.google.com/file/d/1JGLLiu0yIrr_22RLF5Q4dwE_l2MOzVFN/view?usp=drive_link)
>
> The second result for the “loop” scenario:
>
> [Experiment results 5: anonymous link](https://drive.google.com/file/d/119dxJfP5XnbfpQmoPCB4WFUu653o6jTq/view?usp=drive_link)
>
> The result obtained by our algorithm is much more accurate than the result of METIS .

---

> > ### Author Response · Authors · 2024-11-19
> > **Answer to Reviewer hygj Part 3**
> >
> > **Q1. “What would be the potential pros and cons compared to the classical methods, e.g., adaptive mesh refinement? The mesh coarsening task would be interesting, but the reviewer is not sure if this direction is promising, given that we have plenty of optimized methods in the classical numerical analysis field.”**
> >
> > Answer for Q1. Thank you for your thoughtful comment regarding comparing classical methods like adaptive mesh refinement (AMR). While we recognize the robustness of AMR in the field of numerical analysis, our proposed approach has several unique advantages that we believe make it a valuable contribution:
> >
> > First, our algorithm's goal is to reduce the number of grid points while preserving critical solution characteristics, e.g., maintaining accuracy at measurement points or sinks. For example, measurement points could be located near wells in petroleum tasks. Coarsening is designed to optimize the mesh to speed-up the simulation, our goal is not to adapt the mesh but to accelerate the simulation.
> > Our pipeline is fully differentiable, allowing gradients to flow through the coarsening step and the solver. We expect our framework to allow the community to apply different loss functions such as topological and physical losses which is easy since our framework is written in pytorch and JAX.
> > For demonstration, we designed a custom error indicator based on RMSE between fine and coarse solutions at sinks. Just for demonstration purposes, we used the FEniCS FEM package to implement this, but the principle applies to FVM workflows. While we can initially specify the number of points (or resolution) in the coarse grid, the algorithm dynamically increases the number of points in regions where the error indicator exceeds a predefined threshold. Below we present results on how AMR works when we design custom error indicator based on RMSE at sinks (however, AMR typically uses different error indicators). Therefore, The AMR needs to distinguish, based on some heuristics, where to refine and where not to refine. Our algorithm, based on the data from the several measurement points, independently decides how best to arrange all grid points.
> >
> > [Experiment result: anonymous link](https://drive.google.com/file/d/1fB9DJa3dPhDRCvVRsmMEkzzSug-pT_85/view?usp=drive_link)
> >
> > AMR is more commonly applied to FEM due to its simpler refinement criteria. In FVM, ensuring flux conservation during dynamic refinement adds significant complexity, making AMR tools less general-purpose. Our method provides a flexible alternative for coarsening FVM grids while respecting conservation laws and ensuring accuracy at physically meaningful points.

---

> > > ### Author Response · Authors · 2024-11-22
> > > **Answer to Reviewer hygj Part 4**
> > >
> > > We hope the detailed explanations and additional experiments provided have addressed your concerns. If you have any further questions or issues, please feel free to reach out to us, and we would be more than happy to assist. Otherwise, we kindly ask for your support in raising the manuscript's score. We will incorporate all the provided explanations in the original script.

---

> > > > ### Comment · Reviewer_hygj · 2024-11-26
> > > >
> > > > Thank you for the careful rebuttal. Overall, my essential concerns remain the same; thus, I keep the score as is.
> > > >
> > > > In particular, the following two points are not addressed.
> > > >
> > > > * The experimental evaluation is weak.
> > > >   * The authors pointed out differences between the proposed method and the ones raised by the reviewer, but that does not ground that the comparison is impossible. Without comparing more existing methods, the evaluation remains weak.
> > > > * The effectiveness of the proposed method is limited.
> > > >   * I am not sure if I understand the additional results because experimental conditions were not clearly explained; it still seems the method does not have practical superiority.
> > > >   * This kind of speedup is always a tradeoff against accuracy. So, as I wrote in the review, the authors should show a comparison of a speed-accuracy tradeoff rather than a comparison on one hyperparameter setting.
> > > >   * Again, the authors did not show the prediction time of the proposed method, which is essential for the application domain.
> > > >
> > > > I raise some minor points as follows:
> > > >
> > > > > we have claimed an implicit differentiable Finite Volume solver to be used in our coarsening framework.
> > > >
> > > > In that case, an FVM solver build on differentiable framework (pytorch) also exists ( https://openreview.net/forum?id=WajJf47TUi ).
> > > >
> > > > > The term “implicit” in this paper means using implicit updating scheme for message passing instead of a classical explicit updating message passing mechanism. This is completely different from our “implicit” meaning in the context of implicit scheme for solving linear systems.
> > > >
> > > > I do not agree with that. Implicit GNNs solve the (in general nonlinear) algebraic equation, and so does the implicit method in numerical analysis. Hence, those two have a strong relationship.

---

> ### Author Response · Authors · 2024-12-02
> **Additional experiments and remarks**
>
> Dear Reviewer hygj. Thank you for your comment! We have added new experiments and comparison of our algorithm with MeshGraphNets and FluxGNN. Also we have thoroughly updated our paper. See our general replies. Experiment shows that such surrogate-based approaches as MeshGraphNets, even if we use same data, same timestep for training and same timestep for inference as our method, demonstrate worse accuracy in inference even on fine grid. But our method is more data-efficient and our simulator works on coarse grid (obviosly, because our algorithm is coarsening algorithm).  Additionally, we checked FluxGNN and their code is irreproducible (see also our general reply).
>
> We thank you for thoughtful comment and agree that implicit GNNs and implicit numerical methods share a conceptual foundation, as both involve solving algebraic equations iteratively. While the goals and areas of application are somewhat different, it would indeed be interesting to explore the possibility of designing a surrogate solver based on implicit GNNs in future work. However, in this study, we focus on integrating an exact numerical solver using implicit schemes (e.g., backward Euler) to ensure precise and stable solutions for physical simulations without relying on approximations or surrogate models. Also we developed and used fully diffferentiable Voronoi tessellation.
> We will add a clarification in the camera-ready to highlight this distinction and the unique focus of our work.
>
> Additionally, we will add analysis of speed-accuracy tradeoff in some scenarios while camera-ready period.

---

### Author Response · Authors · 2024-11-25
**Dear Reviewers**

A very important point to note for many concerns is the different nature of [Shumilin et al.] baseline from usual baselines. In the [Shumilin et al.] is solved a very difficult problem of global optimization of mesh points, which is performed for every new grid. In the proposed by us approach the GNN outputs directly a solution of this optimization problem which is much more simple and general. That's why this baseline can be considered as the one we would like to be close with (so it's more about the best achievable metrics than about the metrics we can beat). So we consider a major strength of our approach that it gives results very close to the one by [Shumilin et al.].

---

### Author Response · Authors · 2024-12-02
**Comparison with other methods**

Dear Reviewers,

next, we provide the comparison with other methods as some of you have asked.

**MeshGraphNets** [1]

We compared the *accuracy* of the solution obtained by our algorithm vs MeshGraphNets. The experiment setup is as follows: we generate a complex permeability field as the input of the diffusion initial and boundary value problem. The number of initial points is $N$ = 1000. We used an explicit numerical solver to generate data as long as MeshGraphNets is trained on data. The input solution matrix is $U \in R^{N, T}$, where $T$ is the number of time steps. We took a source code and adjusted it for our diffusion problem. As the input to MGN we used node embeddings as $[x_i, y_i, k_i, p^t_i]$  - coordinates, permeability, pressure at the node $i$ at time $t$. We added some noise for $k_i, p^t_i$. For edge embeddings, we used the technique from the original paper: for $e_{ij}$ we embed $ [ \Delta x. \Delta y, ||x_i - x_j|| ] $.  As the output of the graph neural network with 8 hidden layers (k-hop size) we predicted $\partial u/ \partial t$ and then integrated it to get $u_{t+1}$. The timestep is 5e-5. We calculated the roll-out of the trained model in an auto-regressive manner: $X^{t+1} = MeshGraphNets(X^{t})$.
We trained our method on the same grid. We located a source point in the center (as in the original dataset) and two sink (measurement, interest) points. After we applied our method and got a coarsened grid we compared the solution of the rollout and our method's solution at the sink points. The important point to note: MGN and Our method shared the same graph for GNN. The following are the results of the comparison:

[Result of coarsening](https://drive.google.com/file/d/1LaUAYD3k9nE2fKe3raoas6swh7LS2iHJ/view?usp=sharing)

[Comparison at the sink point 1 with MGN](https://drive.google.com/file/d/1_rufwFls9H_UuTC18WaWFDDFHxkShREc/view?usp=sharing)

[Comparison at the sink point 2 with MGN](https://drive.google.com/file/d/1b2wP_3Rnx_nKClzKfHacYrUYPRBFKAyf/view?usp=sharing)

Our method is much more accurate at the points of interest. Another advantage of our method is that it is faster to make a "rollout" because we leverage a differentiable numerical solver on the already coarsened grid. In contrast, MGN uses a heavy trained GNN, which inevitably increases the time of inference.

After that, we tested the *generalizability* of both methods. To do this, we added significant Gaussian noise to the initial permeability field. And performed inference with both trained models. Our method demonstrated to be much more robust to the added noise. The following are the results of the comparison:

[Comparison of generalizability the sink point 1 with MGN](https://drive.google.com/file/d/1IzPfNy2lW5fLNQzYtDCTvSoDYlPGxtpN/view?usp=sharing)

[Comparison of generalizability at the sink point 2 with MGN](https://drive.google.com/file/d/1dBwh9DhFMEUYrKN68Txpb0JRzeh0aguC/view?usp=sharing)



**FluxGNN** [2]

We used the source code from the repository authors provided and after lots of attempts were unable to install the code of the original paper due to errors related to code dependencies. The code is irreproducible. However, we will mention this work in the camera-ready version.

**Adaptive mesh refinement**

We provided a thorough discussion on this topic supported by experiments in the replies to Reviewer hygj.

**Shumilin 2024** [3]

The closest work which we compared with in detail throughout the paper and rebuttals.

Also in the course of the rebuttal, we compared our work with some other methods. We already updated our paper with the Related work subsection. We will include the above comparisons in the camera-ready version of our paper.


[1] https://arxiv.org/pdf/2010.03409 (LEARNING MESH-BASED SIMULATION WITH GRAPH NETWORKS)

[2] https://openreview.net/forum?id=WajJf47TUi

[3] https://openreview.net/forum?id=kMBvZ40Iu9

---

### Author Response · Authors · 2024-12-02
**Final general reply**

We would like to sincerely thank all the reviewers for their constructive feedback and detailed evaluations of our work. We appreciate the time and effort each of you dedicated to reviewing our manuscript. Below, we highlight the strengths noted by the reviewers and describe the changes made to address your concerns, as well as our plans for additional improvements in the camera-ready version.

- Automatic optimization of the grid with the help of GNN, learned in a self-supervised way, saving human efforts and representing a significant advancement over traditional methods (**wVwb, WkfS, vcgH**).
- Relatively novel mesh coarsening problem (**hygj**).
- Designed stability loss term  (**wVwb**).
- Clear and easy-to-follow explanation  (**hygj**).
- Results are comparable with recent methods based on costly global optimization (**wVwb**).

**Changes made in the revised version**

We have made several improvements to the manuscript in response to the reviewers' feedback. The changes address specific concerns raised and aim to strengthen the clarity and comprehensiveness of the paper.

- Improved formulation of the third contribution as recommended by Reviewer **hygj** (**Introduction**).
- Added "Related Work" paragraph comparing our approach with recent works such as Graph Element Networks and MeshGraphNets. This was recommended by Reviewers **hygj** and **vcgH** (**Section 2.2**).
- Improved the description of the neural network architecture used for coarsening, as recommended by Reviewer **WkfS** (**Section 3.2**).
- Explained the rationale for preserving key points as recommended by Reviewer **WkfS**. Additionally, in Appendix C, we described how to handle cases where sensor locations do not align with grid points (**Section 3.2** and new **Appendix C**).
- Introduced the meaning of stability loss upon its first mention and added a link to its full definition, as recommended by Reviewer WkfS (Section 3.2).
- Described how to adapt our framework to 3D meshes or non-simplex grids, as recommended by Reviewer **WkfS** (**Section 3.2** and new **Appendix B**).
- Clarified the multi-step optimization process and stability loss function, as recommended by Reviewer **WkfS** (**Section 3.4**).
- Added a baseline METIS and incorporated new experiments, as discussed with Reviewer **hygj** (Sections **4.1**, **4.2**, **4.3**, and new **Appendix K**).
- Included experiments for the Proxy model scenario, see discussions with Reviewer **wVwb** (**Section 4.2** and new **Appendix J**).
- Added experiments with more complex permeability scenarios (log-uniform permeability), as recommended by Reviewer **wVwb** (**Section 4.5**).
- Provided motivation for using implicit solvers, as recommended by Reviewer **WkfS** (**Appendix M**).
- Incorporated a better set of hyperparameters, as discussed in our response to Reviewer **hygj** (**Appendices I.1 and I.2**).
- Moved time and memory tests to **Appendix H** for better clarity.
- Moved the formal definition of Graph Coarsening for physics-based simulations to **Appendix A**.

**Additional experiments for the camera-ready version**

We are planning further improvements for the camera-ready version, which include:

- Adding experiments comparing our algorithm with the baseline **MeshGraphNets**.
- Mentioning other works such as Universal Physics Transformers, Transolver and FluxGNN.
- Incorporating the remaining textual improvement suggestions by Reviewer **vcgH** into the manuscript.

Once again, we sincerely thank all reviewers for their thoughtful feedback, which has significantly improved the quality of our work. We are especially grateful to **wVwb** and **WkfS** for raising their scores, and we hope all reviewers will consider **raising their scores further**, taking into account the extensive improvements made to the paper and the additional experiments included.

Your insights have been invaluable, and we look forward to further refining our work in the camera-ready version.

Best regards,

The authors

---

### Meta-Review · Area_Chair_znjq · 2024-12-20

**Metareview:**

The work proposes a GNN based method for coarsening the mesh used in a physical simulation, promoting stability and accuracy of the solution. This can speed up traditional numerical methods as much fewer computational nodes can be used to a keep a stable solution. The method is evaluated a few relatively simply PDEs with simple domains.

**Additional Comments On Reviewer Discussion:**

While I generally think that works in this direction are very good for the field, I share the reviewers' concerns about the limited demonstrations and benchmarks in the current work. Numerics on more complicated PDEs with complex domains would really show the applicability of the method. It is quite hard to justify when all domains are boxes.

---

### Decision · Program_Chairs · 2025-01-22

Reject